# Zero-shot Model-based Reinforcement Learning using Large Language Models

**Abdelhakim Benechehab**[†12]**, Youssef Attia El Hili**[1]**, Ambroise Odonnat**[13]**, Oussama Zekri**[‡4]**,
Albert Thomas**[1]**, Giuseppe Paolo**[1]**, Maurizio Filippone**[5]**, Ievgen Redko**[1]**, Balázs Kégl**[1]

[1] Huawei Noah's Ark Lab, Paris, France
[2] Department of Data Science, EURECOM
[3] Inria, Univ. Rennes 2, CNRS, IRISA
[4] ENS Paris-Saclay
[5] Statistics Program, KAUST

## Abstract

The emerging zero-shot capabilities of Large Language Models (LLMs) have led to their applications in areas extending well beyond natural language processing tasks. In reinforcement learning, while LLMs have been extensively used in text-based environments, their integration with continuous state spaces remains understudied. In this paper, we investigate how pre-trained LLMs can be leveraged to predict in context the dynamics of continuous Markov decision processes. We identify handling multivariate data and incorporating the control signal as key challenges that limit the potential of LLMs' deployment in this setup and propose Disentangled In-Context Learning (DICL) to address them. We present proof-of-concept applications in two reinforcement learning settings: model-based policy evaluation and data-augmented off-policy reinforcement learning, supported by theoretical analysis of the proposed methods. Our experiments further demonstrate that our approach produces well-calibrated uncertainty estimates. We release the code at https://github.com/abenechehab/dicl.

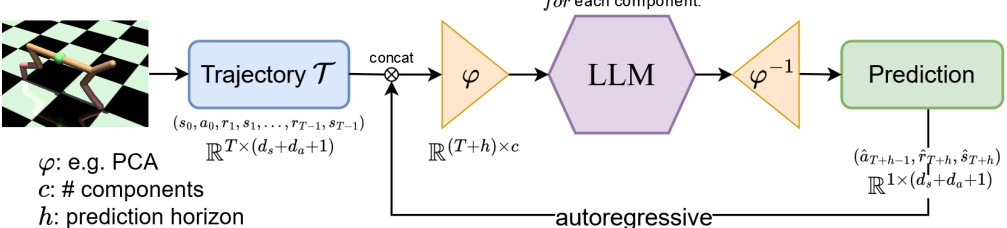

Figure 1: **The DICL Framework.** DICL projects trajectories into a disentangled feature space before performing zero-shot forecasting using a pre-trained LLM and in-context learning.

## 1 Introduction

The rise of large language models (LLMs) has significantly impacted the field of Natural Language Processing (NLP). LLMs (Brown et al., 2020; Hugo Touvron & the Llama 2 team., 2023; Dubey & the Llama 3 team., 2024), which are based on the transformer architecture (Vaswani et al., 2017), have redefined tasks such as machine translation (Brown et al., 2020), sentiment analysis (Zhang et al., 2023b), and question answering (Roberts et al., 2020; Pourkamali & Sharifi, 2024) by enabling machines to understand and generate human-like text with remarkable fluency. One of the most intriguing aspects of LLMs is their emerging capabilities, particularly in-context learning (ICL) (von Oswald et al., 2023). Through ICL, an LLM can learn to perform a new task simply by being provided examples of the task within its input context, without any gradient-based optimization. This

---

[†]Correspondence to abdelhakim.benechehab@gmail.com. [‡]Work done while at Huawei Noah's Ark Lab.

phenomenon has been observed not only in text generation but also in tasks such as image classification (Abdelhamed et al., 2024; Zheng et al., 2024) and even solving logic puzzles (Giadikiaroglou et al., 2024), which is unexpected in the context of the standard statistical learning theory. To our knowledge, ICL capabilities of pre-trained LLMs have been only scarcely explored in reinforcement learning (Wang et al., 2023) despite the demonstrated success of the former in understanding the behavior of deterministic and chaotic dynamical systems (Liu et al., 2024c).

In this paper, we show how ICL with pre-trained LLMs can improve the sample efficiency of Reinforcement Learning (RL), with two proof-of-concepts in policy evaluation and data-augmented off-policy RL. Following the dynamical system perspective on ICL introduced in Li et al. (2023) and experimentally studied in Liu et al. (2024c), we use the observed trajectories of a given agent to predict its future state and reward in commonly used RL environments. To achieve this, we solve two crucial challenges related to considering continuous state-space Markov Decision Processes (MDP): 1) incorporating the action information into the LLM's context and 2) handling the interdependence between the state-actions dimensions, as prior approaches were known to treat multivariate data's covariates independently. Our framework, DICL (Disentangled In-Context Learning), is summarized in Fig. 1. The core idea of DICL is to apply a feature space transformation, denoted as $\varphi$, which captures the interdependencies between state and action features in order to disentangle each dimension. Subsequently, a Large Language Model (LLM) is employed to forecast each component independently in a zero-shot manner through in-context learning. Finally, the predictions are transformed back to the original trajectory space using the inverse transformation $\varphi^{-1}$.

Our approach leads to several novel insights and contributions, which we summarize as follows:

1. *Methodological.* We develop a novel approach to integrate state dimension interdependence and action information into in-context trajectories. This approach, termed Disentangled In-Context Learning (DICL), leads to a new methodology for applying ICL in RL environments with continuous state spaces. We validate our proposed approach on tasks involving proprioceptive control.

2. *Theoretical.* We theoretically analyze the policy evaluation algorithm resulting from multi-branch rollouts with the LLM-based dynamics model, leading to a novel return bound.

3. *Experimental.* We show how the LLM's MDP modeling ability can benefit two RL applications: policy evaluation and data-augmented offline RL. Furthermore, we show that the LLM is a calibrated uncertainty estimator, a desirable property for MBRL algorithms.

**Organization of the paper.** The paper is structured as follows: Section 2 introduces the main concepts from the literature used in our work (while a more detailed related work is deferred to Appendix B). We then start our analysis in Section 3.1, by analyzing LLM's attention matrices. DICL is presented in Section 3.3, while Section 4 contains different applications of the proposed method in RL, along with the corresponding theoretical analysis. Finally, Section 5 provides a short discussion and future research directions triggered by our approach.

## 2 BACKGROUND KNOWLEDGE

**Reinforcement Learning (RL).** The standard framework of RL is the infinite-horizon **Markov decision process (MDP)** $\mathcal{M} = \langle \mathcal{S}, \mathcal{A}, P, r, \mu_0, \gamma \rangle$ where $\mathcal{S}$ represents the state space, $\mathcal{A}$ the action space, $P : \mathcal{S} \times \mathcal{A} \to \mathcal{S}$ the (possibly stochastic) transition dynamics, $r : \mathcal{S} \times \mathcal{A} \to \mathbb{R}$ the reward function, $\mu_0$ the initial state distribution, and $\gamma \in [0, 1]$ the discount factor. The goal of RL is to find, for each state $s \in \mathcal{S}$, a distribution $\pi(s)$ over the action space $\mathcal{A}$, called the *policy*, that maximizes the expected sum of discounted rewards $\eta(\pi) := \mathbb{E}_{s_0 \sim \mu_0, a_t \sim \pi, s_{t>0} \sim P^t}[\sum_{t=0}^{\infty} \gamma^t r(s_t, a_t)]$. Under a policy $\pi$, we define the state value function at $s \in \mathcal{S}$ as the expected sum of discounted rewards, starting from the state $s$, and following the policy $\pi$ afterwards until termination: $V^{\pi}(s) = \mathbb{E}_{a_t \sim \pi, s_{t>0} \sim P^t}[\sum_{t=0}^{\infty} \gamma^t r(s_t, a_t) \mid s_0 = s]$.

**Model-based RL (MBRL).** MBRL algorithms address the supervised learning problem of estimating the dynamics of the environment $\hat{P}$ (and sometimes also the reward function $\hat{r}$) from data collected when interacting with the real system. The model's loss function is typically the log-likelihood $\mathcal{L}(\mathcal{D}; \hat{P}) = \frac{1}{N} \sum_{i=1}^{N} \log \hat{P}(s_{t+1}^i | s_t^i, a_t^i)$ or Mean Squared Error (MSE) for deterministic models. The learned model can subsequently be used for policy search under the MDP

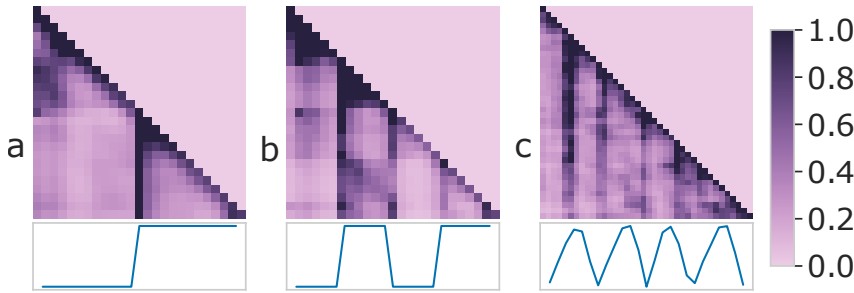

Figure 2: **LLM can perceive time patterns.** The LLM (*Llama 3-8B*) is fed with 3 time series presenting distinct patterns. **(a)** Rectangular pulse. **(b)** Rectangular signal with constant sub-parts. **(c)** The *fthigh* dimension of HalfCheetah under an expert policy. Tokens belonging to constant slots (or peaks) attend to all the similar ones that precede them, focusing more on their first occurrence.

$\widehat{\mathcal{M}} = \langle \mathcal{S}, \mathcal{A}, \hat{P}, r, \mu_0, \gamma \rangle$. This MDP shares the state and action spaces $\mathcal{S}, \mathcal{A}$, reward function $r$, with the true environment $\mathcal{M}$, but learns the transition probability $\hat{P}$ from the dataset $\mathcal{D}$.

**Large Language Models (LLMs).** Within the field of Natural Language Processing, Large Language Models (LLMs) have emerged as a powerful tool for understanding and generating human-like text. An LLM is typically defined as a neural network model, often based on the transformer architecture (Vaswani et al., 2017), that is trained on a vast corpus of sequences, $U = \{U_1, U_2, \ldots, U_i, \ldots, U_N\}$, where each sequence $U_i = (u_1, u_2, \ldots, u_j, \ldots, u_{n_i})$ consists of tokens $u_j$ from a vocabulary $\mathcal{V}$. Decoder-only LLMs (Radford et al., 2019; Dubey & the Llama 3 team., 2024) typically encode an autoregressive distribution, where the probability of each token is conditioned only on the previous tokens in the sequence, expressed as $p_\theta(U_i) = \prod_{j=1}^{n_i} p_\theta(u_j|u_{0:j-1})$. The parameters $\theta$ are learned by maximizing the probability of the entire dataset, $p_\theta(U) = \prod_{i=1}^{N} p_\theta(U_i)$. Every LLM has an associated tokenizer, which breaks an input string into a sequence of tokens, each belonging to $\mathcal{V}$.

**In-Context Learning (ICL).** In order to use trajectories as inputs in ICL, we use the tokenization of time series proposed in Gruver et al. (2023b) and Jin et al. (2024). This approach uses a subset of the LLM vocabulary $\mathcal{V}_{num}$ representing digits to tokenize the time series (Algo-

---

**Algorithm 1** ICL$_\theta$ (Liu et al., 2024b; Gruver et al., 2023b)

**Input:** Time series $(x_i)_{i \leq t}$, LLM $p_\theta$, sub-vocabulary $\mathcal{V}_{num}$
**1.** Tokenize time series $\hat{x}_t = $ "$x_1^1 x_1^2 \ldots x_1^k, \ldots$"
**2.** logits $\leftarrow p_\theta(\hat{x}_t)$
**3.** $\{P(X_{i+1}|x_i, \ldots, x_0)\}_{i \leq t} \leftarrow$ softmax(logits($\mathcal{V}_{num}$))
**Return:** $\{P(X_{i+1}|x_i, \ldots, x_0)\}_{i \leq t}$

---

rithm 1). Specifically, given an univarite time series, we rescale it into a specific range (Liu et al., 2024b; Zekri et al.; Requeima et al., 2024), encode it with $k$ digits, and concatenate each value to build the LLM prompt:

$$\underbrace{[0.2513, 5.2387, 9.7889]}_{\text{time series}} \rightarrow \underbrace{[1.5, 5.16, 8.5]}_{\text{rescaled}} \rightarrow \underbrace{\text{``150, 516, 850''}}_{\text{prompt}}$$

After the LLM forward pass, the logits corresponding to tokens in $\mathcal{V}_{num}$ can be used to predict a categorical distribution over the next value as demonstrated in Liu et al. (2024c), thereby enabling uncertainty estimation.

## 3 ZERO-SHOT DYNAMICS LEARNING USING LARGE LANGUAGE MODELS

### 3.1 MOTIVATION

Before considering the multivariate trajectories of agents collected in RL environments, we first want to verify whether a pre-trained LLM model is sensitive to the primitive univariate signals akin to those encountered in them. For this, we investigate the attention mechanism of the Llama3 8B model (Dubey & the Llama 3 team., 2024) when we feed it with different signals, including the

periodic *fthigh* dimension from the HalfCheetah system (Brockman et al., 2016). By averaging the attention matrices over the 32 heads for each of the 32 layers of the multi-head attention in Llama3, we observed distinct patterns that provide insight into the model's focus and behavior (Fig. 2 shows selected attention layers for each signal). The attention matrices exhibit a diagonal pattern, indicative of strong self-correlation among timestamps, and a subtriangular structure due to the causal masked attention in decoder-only transformers.

Further examination of the attention matrices reveals a more intricate finding. Tokens within repeating patterns (e.g., signal peaks, constant parts) not only attend to past tokens within the same cycle but also to those from previous occurrences of the same pattern, demonstrating a form of in-context learning. The ability to detect and exploit repeating patterns within such signals is especially valuable in RL, where state transitions and action outcomes often exhibit cyclical or recurring dynamics, particularly in continuous control tasks. However, applying this insight to RL presents two critical challenges related to 1) the integration of actions into the forecasting process, and 2) handling of the multivariate nature of RL problems. We now address these challenges by building on the insights from the analysis presented above.

## 3.2 PROBLEM SETUP

Given an initial trajectory $\mathcal{T} = (s_0, a_0, r_1, s_1, a_1, r_2, s_2, \ldots, r_{T-1}, s_{T-1})$ of length $T$, with $s_t \in \mathcal{S}$, $a_t = \pi(s_t) \in \mathcal{A}^\dagger$, where the policy $\pi$ is fixed for the whole trajectory, and $r_t \in \mathbb{R}$, we want to predict future transitions: given $(s_{T-1}, a_{T-1})$ predict the next state and reward $(s_T, r_T)$ and subsequent transitions autoregressively. For simplicity we first omit the actions and the reward, focusing instead on the multivariate sequence $\tau^\pi = (s_0, s_1, \ldots, s_T)$ where we assume that the state dimensions are independent. Later, we show how to relax the assumptions of omitting actions and rewards, as well as state independence, which is crucial for applications in RL. The joint probability density function of $\tau^\pi$ can be written as:

$$\begin{cases} \mathbf{P}(\tau^\pi) = \mu_0(s_0) \prod_{t=1}^{T} P^\pi(s_t|s_{t-1}) \\ \text{where } P^\pi(s_t|s_{t-1}) = \int_{a \in \mathcal{A}} \pi(a|s_{t-1}) P(s_t|s_{t-1}, a)\, da \ . \end{cases} \quad (1)$$

Using the decoder-only nature of the in-context learner defined in Section 2, we can apply Algorithm 1 to each dimension of the state vector to infer the transition rule of each visited state in $\tau^\pi$ conditioned on its relative history: for all $j \in \{1, \ldots, d_s\}$,

$$\{\hat{P}_\theta^{\pi,j}(s_t^j|s_{t-1}^j, \ldots, s_1^j, s_0^j)\}_{t \leq T} = \text{ICL}_\theta(\tau^{\pi,j}) \quad (2)$$

where $\theta$ are the fixed parameters of the LLM used as an in-context learner, and $T$ its context length. Assuming complete observability of the MDP state, the Markovian property unveils an equivalence between the learned transition rules and the corresponding Markovian ones: $\hat{P}_\theta(s_t|s_{t-1}, \ldots, s_1, s_0) = \hat{P}_\theta(s_t|s_{t-1})$.

This approach, that we name vICL (for vanilla ICL), thus applies Algorithm 1 on each dimension of the state individually, assuming their independence. Furthermore, the

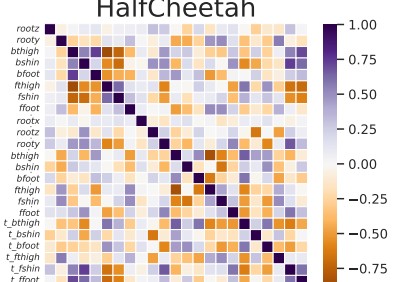

Figure 3: The covariance matrix from an expert dataset in the Halfcheetah environment indicates linear correlations between state-action features.

action information is integrated-out (as depicted in Eq. (1)), which in theory, limits the application scope of this method to quantities that only depend on a policy through the expectation over actions (e.g., the value function $V^\pi(s)$). We address these limitations in the next section.

**On the zero-shot nature of DICL.** Our use of the term "zero-shot" aligns with the literature on LLMs and time series (Gruver et al., 2023a), indicating that we do not perform any gradient updates or fine-tuning of the pretrained LLM's weights. Specifically, we adopt the dynamical systems formulation of ICL as studied in Li et al. (2023), where the query consists of the trajectory "$s_0^j, s_1^j, \ldots, s_{t-1}^j$" and the label is the subsequent value $s_t^j$.

---

$^\dagger$In practice, states and actions are real valued vectors spanning a space of dimensions respectively $d_s$ and $d_a$: $\mathcal{S} = \mathbb{R}^{d_s}, \mathcal{A} = \mathbb{R}^{d_a}$

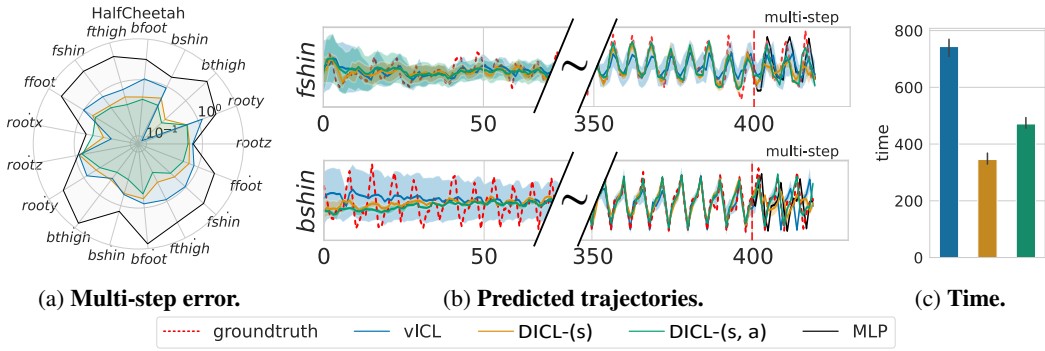

(a) **Multi-step error.**        (b) **Predicted trajectories.**        (c) **Time.**

········· groundtruth  ——— vICL  ——— DICL-(s)  ——— DICL-(s, a)  ——— MLP

Figure 4: **PCA-based DICL achieves smaller multi-step error in less computational time.** We compare DICL-$(s)$ and DICL-$(s, a)$ using a number of components equal to half the number of features, with the vanilla approach vICL and an MLP baseline. (*Llama 3-8B*).

### 3.3 STATE AND ACTION DIMENSION INTERDEPENDENCE

In this section we address the two limitations of vICL discussed in Section 3.2 by introducing Disentangled In-Context Learning (DICL), a method that relaxes the assumption of state feature independence and reintroduces the action by employing strategies that aim to map the state-action vector to a latent space where the features are independent. We can then apply vICL, which operates under the assumption of feature independence, to the latent representation. An added benefit of using such a latent space is that it can potentially reduce the dimensionality, leading to a speed-up of the overall approach.

While sophisticated approaches[†] like disentangled autoencoders could be considered for DICL, in this work we employ Principal Component Analysis (PCA). In fact, the absence of pre-trained models for this type of representation learning requires training from scratch on a potentially large dataset. This goes against our goal of leveraging the pre-trained knowledge of LLMs and ICL. Instead, we find that PCA, which generates new linearly uncorrelated features and can reduce dimensionality, strikes a good balance between simplicity, tractability, and performance (Fig. 3 and Fig. 4). Nonetheless, DICL is agnostic to this aspect and any transformation $\varphi$ that can disentangle features can be used in place of PCA. In the rest of the paper we present two variants of DICL:

- DICL-$(s, a)$, which applies the rotation matrix of PCA to the feature space of states and actions and then runs Algorithm 1 in the projection space of principal components;

- DICL-$(s)$, which applies the same transformation solely to the trajectory of states. This is useful in settings in which integrating the actions is not necessary, as when we only want to estimate the value function $V^\pi(s)$.

### 3.4 AN ILLUSTRATIVE EXAMPLE

In this section, we aim to challenge our approach against the HalfCheetah system from the MuJoCo Gym environment suite (Brockman et al., 2016; Todorov et al., 2012). All our experiments are conducted using the Llama 3 series of models (Dubey & the Llama 3 team., 2024). Fig. 4a shows the average MSE over a prediction horizon of $h \in \{1, \ldots, 20\}$ steps for each state dimension. Fig. 4b shows predicted trajectories for selected state dimensions of the HalfCheetah system (the details of the experiment, the metrics and the remaining state dimensions are deferred to Appendix F).

We first observe that the LLM-based dynamics forecasters exhibit a burn-in phase ($\approx 70$ steps in Fig. 4b) that is necessary for the LLM to gather enough context. For multi-step prediction, Fig. 4a, showing the average MSE over prediction horizons and trajectories, demonstrates that both versions of DICL improve over the vanilla approach and the MLP baseline trained on the context data, in almost all state dimensions. Indeed, we hypothesize that this improvement is especially brought by the projection in a linearly uncorrelated space that PCA enables. Furthermore, we also leveraged the

---

[†]A more detailed discussion of alternative approaches to PCA is provided in Appendix C.

dimensionality reduction feature by selecting a number of components $c$ equal to half the number of the original features $d_s + d_a$ (or $d_s$ in DICL-$(s)$). This results in a significant decrease in the computational time of the method without loss of performance, as showcased by Fig. 4c.

**LLMs comparison.** In Table 1 we compare the performance obtained by the baselines and DICL when using different LLMs. Similarly to Fig. 4a, the scores are calculated as the average over a given prediction horizon $h$ across all dimensions (refer to Appendix F for details on the MSE, and Appendix G for details on the KS statistic). Note that similarly to Fig. 4, we use PCA-based dimensionality reduction for both DICL-$(s, a)$ and DICL-$(s)$ in this experiment, *reducing the original number of features by half*. Overall, we can see that DICL, especially the DICL-$(s, a)$ version, demonstrates improved calibration compared to both vICL and the MLP baselines, thanks to the disentangling effect of PCA. Moreover, DICL-$(s)$ with the 3.1-70B model achieves the lowest Mean Squared Error (MSE) of 3.59. Nonetheless, DICL-$(s, a)$ exhibits the highest MSE across all models. This is likely due to the additional error introduced by predicting action information, thereby modeling both the dynamics and the data-generating policy. This aspect differs from the MLP baseline, which is provided with real actions at test time (acting as an oracle), and from DICL-$(s)$ and vICL, which operate solely on states. We show the detailed results of this ablation study in Appendix H. Notice that we exclusively used LLMs based on the LLaMA series of models (Dubey & the Llama 3 team., 2024). This was a strategic choice due to the LLaMA tokenizer, which facilitates our framework by assigning a separate token to each number between 0 and 999. For other LLMs, algorithms have been suggested in the literature to extract transition rules from their output logits. For example, the Hierarchical Softmax algorithm (Liu et al., 2024b) could be employed for this purpose.

| LLaMA | Metrics | |
|---|---|---|
| | $\mathbf{MSE}/10^{-2}\downarrow$ | $\mathbf{KS}/10^{-2}\downarrow$ |
| **vICL** | | |
| 3.2-1B | $384 \pm 31$ | $52 \pm 7$ |
| 3.2-3B | $399 \pm 40$ | $54 \pm 8$ |
| 3.1-8B | $380 \pm 32$ | $53 \pm 7$ |
| 3-8B | $375 \pm 30$ | $53 \pm 7$ |
| 3.1-70B | $392 \pm 35$ | $55 \pm 7$ |
| **DICL-$(s)$** | | |
| 3.2-1B | $389 \pm 38$ | $50 \pm 7$ |
| 3.2-3B | $404 \pm 41$ | $51 \pm 7$ |
| 3.1-8B | $372 \pm 44$ | $50 \pm 7$ |
| 3-8B | $370 \pm 36$ | $50 \pm 7$ |
| 3.1-70B | $\mathbf{359 \pm 33}$ | $54 \pm 7$ |
| **DICL-$(s, a)$** | | |
| 3.2-1B | $449 \pm 37$ | $46 \pm 5$ |
| 3.2-3B | $450 \pm 47$ | $48 \pm 6$ |
| 3.1-8B | $412 \pm 39$ | $\mathbf{45 \pm 6}$ |
| 3-8B | $418 \pm 46$ | $46 \pm 5$ |
| 3.1-70B | $428 \pm 47$ | $47 \pm 5$ |
| **baseline** | | |
| MLP | $406 \pm 59$ | $55 \pm 3$ |

Table 1: Comparison of different LLMs. Results are average over 5 episodes from each one of 7 D4RL (Fu et al., 2021) tasks. ↓ means lower the better. The best average score is shown in **bold**. We show the average score $\pm$ the 95% Gaussian confidence interval.

## 4 USE-CASES IN REINFORCEMENT LEARNING

As explored in the preceding sections, LLMs can be used as accurate dynamics learners for proprioceptive control through in-context learning. We now state our main contributions in terms of the integration of DICL into MBRL. First, we generalize the return bound of Model-Based Policy Optimization (MBPO) (Janner et al., 2019) to the more general case of multiple branches and use it to analyze our method. Next, we leverage the LLM to augment the replay buffer of an off-policy RL algorithm, leading to a more sample-efficient algorithm. In a second application, we apply our method to predict the reward signal, resulting in a hybrid model-based policy evaluation technique. Finally, we show that the LLM provides calibrated uncertainty estimates and conclude with a discussion of our results.

### 4.1 THEORETICAL ANALYSIS: RETURN BOUND UNDER MULTI-BRANCH ROLLOUTS

When using a dynamics model in MBRL, one ideally seeks monotonic improvement guarantees, ensuring that the optimal policy under the model is also optimal under the true dynamics, up to some bound. Such guarantees generally depend on system parameters (e.g., the discount factor $\gamma$), the prediction horizon $k$, and the model generalization error $\varepsilon_{\mathrm{m}}$. As established in Janner et al. (2019) and Frauenknecht et al. (2024), the framework for deriving these theoretical guarantees is the one of branched model-based rollouts.

A branched rollout return $\eta^{\text{branch}}[\pi]$ of a policy $\pi$ is defined in Janner et al. (2019) as the return of a rollout which begins under the true dynamics $P$ and at some point in time switches to rolling out under learned dynamics $\hat{P}$ for $k$ steps.

For our LLM-based dynamics learner, we are interested in studying a more general branching scheme that will be later used to analyze the results of our data-augmented off-policy algorithm. We begin by defining the multi-branch rollout return.

**Definition 4.1** (Multi-branch rollout return). The multi-branch rollout return $\eta^{\text{llm}}_{p,k,T}[\pi]$ of a policy $\pi$ is defined as the expected return over rollouts with the following dynamics:

1. for $t < T$, where $T$ is the minimal context length, the rollout follows the true dynamics $P$.

2. for $t \geq T$, with probability $p$, the rollout switches to the LLM-based dynamics $\hat{P}_{\text{llm}}$ for $k$ steps, otherwise the rollout continues with the true dynamics $P$.

$$\eta^{\text{llm}}_{p,k,T}[\pi] = \mathbb{E}[branches]$$

Figure 5: **Multi-branch return.** The rollout following the true dynamics $P$ is shown in blue. The branched rollouts following LLM-based dynamics $\hat{P}_{\text{llm}}$ are in purple. Branched rollouts can overlap, with the expectation over the overlapping branches as the return.

These different rollout realizations, referred to as branches, can overlap, meaning that multiple LLM-based dynamics can run in parallel if multiple branchings from the true dynamics occur within the $k$-step window (see Fig. 5).

With this definition, we now state our main theoretical result, consisting of a return bound between the true return and the multi-branch rollout return.

**Theorem 4.2** (Multi-branch return bound). *Let $T$ be the minimal length of the in-context trajectories, $p \in [0, 1]$ the probability that a given state is a branching point. We assume that the reward is bounded and that the expected total variation between the LLM-based model and the true dynamics under a policy $\pi$ is bounded at each timestep by $\max_{t \geq T} \mathbb{E}_{s \sim P^t, a \sim \pi}[D_{TV}(P(.|s,a)||\hat{P}_{llm}(.|s,a))] \leq \varepsilon_{llm}(T)$. Then under a multi-branched rollout scheme with a branch length of $k$, the return is bounded as follows:*

$$|\eta(\pi) - \eta^{llm}_{p,k,T}(\pi)| \leq 2\frac{\gamma^T}{1-\gamma}r_{\max}k^2\,p\,\varepsilon_{llm}(T) \ , \tag{3}$$

*where $r_{\max} = \max_{s \in \mathcal{S}, a \in \mathcal{A}} r(s,a)$.*

Theorem 4.2 generalizes the single-branch return presented in Janner et al. (2019), incorporating an additional factor of the prediction horizon $k$ due to the presence of multiple branches, and directly accounting for the impact of the amount of LLM training data through the branching factor $p$. Additionally, the bound is inversely proportional to the minimal context length $T$, both through the power in the discount factor $\gamma^T$ and the error term $\varepsilon_{\text{llm}}(T)$. Indeed, the term $\varepsilon_{\text{llm}}(T)$ corresponds to the generalization error of in-context learning. Several works in the literature studied it and showed that it typically decreases in $\mathcal{O}(T^{-1/2})$ with $T$ the length of the context trajectories (Zekri et al., 2024; Zhang et al., 2023c; Li et al., 2023).

### 4.2 DATA-AUGMENTED OFF-POLICY REINFORCEMENT LEARNING

In this section, we show how DICL can be used for data augmentation in off-policy model-free RL algorithms such as Soft Actor-Critic (SAC) (Haarnoja et al., 2018). The idea is to augment the replay buffer of the off-policy algorithm with transitions generated by DICL, using trajectories already collected by previous policies. The goal is to improve sample-efficiency and accelerate the learning curve, particularly in the early stages of learning as the LLM can generate accurate transitions from a small trajectory. We name this application of our approach DICL-SAC.

As defined in Corrado & Hanna (2023), data-augmented off-policy RL involves perturbing previously observed transitions to generate new transitions, without further interaction with the environment. The generated transitions should ideally be diverse and feasible under the MDP dynamics to enhance sample efficiency while ensuring that the optimal policy remains learnable.

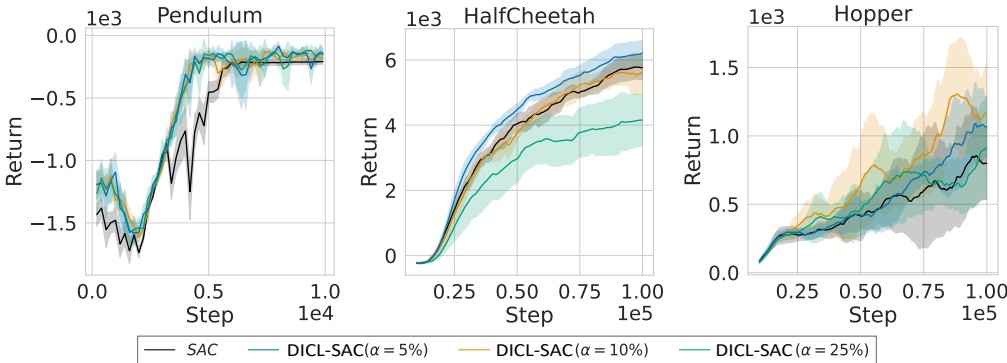

Figure 6: **Data-augmented off-policy RL.** In the early stages of training DICL-SAC improves the sample efficiency of SAC on three Gym control environments. Due to the intensive use of the LLM within DICL-SAC, we conducted this experiment using the *Llama 3.2-1B* model.

Algorithm 2 (DICL-SAC) integrates multiple components to demonstrate a novel proof-of-concept for improving the sample efficiency of SAC using DICL for data augmentation. Let $\mathcal{T} = (s_0, a_0, r_0, \ldots, s_{T_{\max}}, a_{T_{\max}}, r_{T_{\max}})$ be a real trajectory collected with a fixed policy $\pi_\phi$, sampled from the real transitions being stored in a replay buffer $\mathcal{R}$. We generate synthetic transitions $(s_t, \tilde{a}_t, r_t, \hat{s}_{t+1})_{T \leq t \leq T_{\max}}$; where $\hat{s}_{t+1}$ is the next state generated by the LLM model applied on the trajectory of the states only, $\tilde{a}_t$ is an action sampled from the data collection policy $\pi_\phi(.|s_t)$, and $T$ is the minimal context length. These transitions are then stored in a separate replay buffer $\mathcal{R}_{\text{llm}}$. At a given update frequency, DICL-SAC performs $G$ gradient updates using data sampled from $\mathcal{R}$

**Algorithm 2** DICL-SAC

1: **Inputs:** LLM-based dynamics learner (e.g. DICL-$(s)$), batch size $b$, LLM data proportion $\alpha$, minimal context length $T$, and maximal context length $T_{\max}$
2: **Initialize** policy $\pi_\phi$, critic $Q_\psi$, replay buffer $\mathcal{R}$, and LLM replay buffer $\mathcal{R}_{\text{llm}}$, and context size $T_{\max}$
3: **for** $t = 1, \ldots, N\_interactions$ **do**
4:     New transition $(s_t, a_t, r_t, s_{t+1})$ from $\pi_\theta$
5:     Add $(s_t, a_t, r_t, s_{t+1})$ to $\mathcal{R}$
6:     Store auxiliary action $\tilde{a}_t \sim \pi_\theta(.|s_t)$
7:     **if** Generate LLM data **then**
8:         Sample trajectory $\mathcal{T} = (s_0, \ldots, s_{T_{\max}})$ from $\mathcal{R}$
9:         $\{\hat{s}_{i+1}\}_{0 \leq i \leq T_{\max}} \sim$ DICL-$(s)$ $(\mathcal{T})$
10:        Add $\{(s_i, \tilde{\tilde{a}}_i, r_i, \hat{s}_{i+1})\}_{T \leq i \leq T_{\max}}$ to $\mathcal{R}_{\text{llm}}$
11:    **end if**
12:    **if** update SAC **then**
13:        Sample batch $\mathcal{B}$ of size $b$ from $\mathcal{R}$
14:        Sample batch $\mathcal{B}_{\text{llm}}$ of size $\alpha \cdot b$ from $\mathcal{R}_{\text{llm}}$
15:        Update $\phi$ and $\psi$ on $\mathcal{B} \cup \mathcal{B}_{\text{llm}}$
16:    **end if**
17: **end for**

and $\alpha\% \cdot G$ gradient updates using data sampled from $\mathcal{R}_{\text{llm}}$. Other hyperparameters of our method include the LLM-based method (vICL, DICL-$(s)$ or DICL-$(s,a)$), how often we generate new LLM data and the maximal context length $T_{\max}$ (see Appendix D for the full list of hyperparameters).

Fig. 6 compares the return curves obtained by DICL-SAC against SAC in three control environments from the Gym library (Brockman et al., 2016). As anticipated with our data augmentation approach, we observe that our algorithm improves the sample efficiency of SAC at the beginning of training. This improvement is moderate but significant in the Pendulum and HalfCheetah environments, while the return curves tend to be noisier in the Hopper environment. Furthermore, as the proportion of LLM data $\alpha$ increases, the performance of the algorithm decreases (particularly in HalfCheetah), as predicted by Theorem 4.2. Indeed, a larger proportion of LLM data correlates with a higher probability of branching $p$, as more branching points will be sampled throughout the training. Regarding the other parameters of our bound in Theorem 4.2, we set $T = 1$, meaning all LLM-generated transitions are added to $\mathcal{R}_{\text{llm}}$, and $k = 1$ to minimize LLM inference time.

## 4.3 POLICY EVALUATION

In this section we show how DICL can be used for policy evaluation.

System engineers are often presented with several policies to test on their systems. On the one hand, off-policy evaluation (e.g., Uehara et al. (2022)) involves using historical data collected from a different policy to estimate the performance of a target policy without disrupting the system. However, this approach is prone to issues such as distributional shift and high variance. On the other hand, online evaluation provides a direct and unbiased comparison under real conditions. System engineers often prefer online evaluation for a set of pre-selected policies because it offers real-time feedback and ensures that deployment decisions are based on live data, closely reflecting the system's true performance in production. However, online evaluations can be time-consuming and may temporarily impact system performance. To address this, we propose a hybrid approach using LLM dynamics predictions obtained through ICL to reduce the time required for online evaluation: the initial phase of policy evaluation is conducted as a standard online test, while the remainder is completed offline using the dynamics predictions enabled by the LLM's ICL capabilities.

Fig. 7 illustrates the relative error in value obtained by predicting the trajectory of rewards for $k$ steps, given a context length of $T = 500$. When $k \leq 500$, we complete the remaining steps of the 1000-step episode using the actual rewards. For the two versions of DICL, the reward vector is concatenated to the feature space prior to applying PCA. In the Hopper environment, it is evident that predicting the reward trajectory alone is a challenging task for the vanilla method vICL. On the contrary, both DICL-$(s)$ and DICL-$(s, a)$ effectively capture some of the dependencies of the reward signal on the states and actions, providing a more robust method for policy evaluation, and matching the MLP baseline that has been trained on a

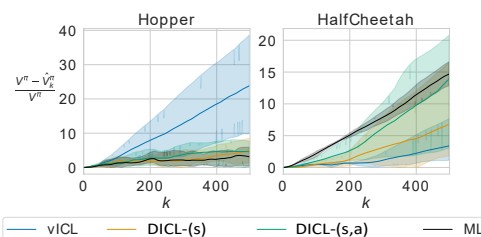

Figure 7: **Policy evaluation with DICL.** Relative error on the predicted value over $k = 500$ steps, with context length of $T = 500$. This experiment is conducted using the *Llama 3-8B* model.

dataset of transitions sampled from the same policy. However, in HalfCheetah we observe that the vanilla method largely improves upon both the baseline and DICL. We suspect that this is due to the fact that the reward signal is strongly correlated with the $rootx$ dimension in HalfCheetah, which proved to be harder to predict by our approach, as can be seen in Fig. 4a.

Note that the experimental setup that we follow here is closely related to the concept of Model-based Value Expansion (Feinberg et al., 2018; Buckman et al., 2018), where we use the dynamics model to improve the value estimates through an n-step expansion in an Actor Critic algorithm.

## 4.4 CALIBRATION OF THE LLM UNCERTAINTY ESTIMATES

An intriguing property observed in Fig. 4b is the confidence interval around the predictions. As detailed in Algorithm 1, one can extract a full probability distribution for the next prediction given the context, enabling uncertainty estimation in the LLM's predictions. Notably, this uncertainty is pronounced at the beginning when context is limited, around peaks, and in regions where the average prediction exhibits large errors. We explore this phenomenon further in the next section by evaluating the calibration of the LLM's uncertainty estimates.

Calibration is known to be an important property of a dynamics model when used in reinforcement learning (Malik et al., 2019). In this section, we aim to investigate whether the uncertainty estimates derived from the LLM's logits are well-calibrated. We achieve this by evaluating the quantile calibration (Kuleshov et al., 2018) of the probability distributions obtained for each LLM-based method.

**Quantile calibration.** For a regression problem with variable $y \in \mathcal{Y} = \mathbb{R}$, and a model that outputs a cumulative distribution function (CDF) $F_i$ over $y_i$ (where $i$ indexes data points), quantile calibration implies that $y_i$ (groundtruth) should fall within a $p\%$-confidence interval $p\%$ of the time:

$$\frac{\sum_{i=1}^{N} \mathbb{I}\{y_i \leq F_i^{-1}(p)\}}{N} \to p \quad \text{for all} \quad p \in [0, 1] \quad \text{as} \quad N \to \infty \tag{4}$$

where $F_i^{-1} : [0, 1] \rightarrow \mathcal{Y}$ denotes the quantile function $F_i^{-1}(p) = \inf\{y : p \leq F_i(y)\}$ for all $p \in [0, 1]$, and $N$ the number of samples.

**LLMs are well-calibrated forecasters.** Fig. 8 shows the reliability diagram for the $bfoot$ dimension of the HalfCheetah system. The overall conclusion is that, regardless of the LLM-based sub-routine used to predict the next state, the uncertainty estimates derived from the LLM's logits are well-calibrated in terms of quantile calibration. Ideally, forecasters should align with the diagonal in Fig. 8, which the LLM approach nearly achieves. Furthermore, when comparing with a naive baseline (the details are deferred to Appendix G), the LLM-forecaster matches the baseline when it's already calibrated, and improves over it when it's not. To quantify a forecaster's calibration with a point statistic, we compute the Kolmogorov-Smirnov goodness-of-fit test Eq. (10), shown in the legend of Fig. 8.

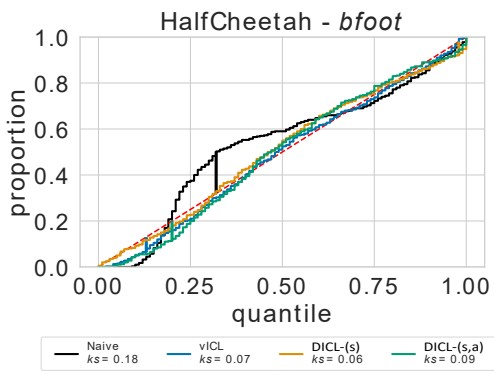

Figure 8: **Quantile calibration reliability diagram.** The LLM (*Llama 3 8B*) uncertainty estimates are well-calibrated. Vertical lines show the Kolmogorov-Smirnov statistic for each fit.

## 5 DISCUSSION

By introducing the DICL framework, our goal is to bridge the gap between MBRL and LLMs. Our study raises multiple open questions and future research directions. Notably, the choice of the feature transformation is crucial for improving performance in specific applications. We plan to explore transformations that capture not only linear but also non-linear dependencies, such as AutoEncoders, as discussed in Appendix C. Another possible direction is the integration of textual context information into the LLM prompt. This approach has been shown to enhance the overall pipeline for time series forecasting (Jin et al., 2024; Xue & Salim, 2023) and policy learning (Wang et al., 2023).

Besides this, our algorithm DICL-SAC performs data augmentation by applying the LLM to generate next states Eq. (2). This operation requires a total of $d_s$ calls to the LLM (or $c$ after the $\varphi$ transformation) to generate $T_{\max} - T$ transitions, as the time steps can be batched. This approach assumes a fixed policy in the context, allowing the LLM to implicitly learn $P^{\pi_\phi}$ using only the states. Looking ahead, a future research direction is to explore how to apply DICL to MBRL by replacing the dynamics model with an LLM. Naively applying DICL-$(s, a)$ would require $(T_{\max} - T) \cdot d_s$ calls to the LLM, as transitions need to be predicted sequentially when actions change. This results in an extremely computationally expensive method, making it infeasible for many applications. Therefore, further research is needed to make this approach computationally efficient.

### CONCLUSION

In this paper, we ask how we can leverage the emerging capabilities of Large Language Models to benefit model-based reinforcement learning. We build on previous work that successfully conceptualized in-context learning for univariate time series prediction, and provide a systematic methodology to apply ICL to an MDP's dynamics learning problem. Our methodology, based on a projection of the data in a linearly uncorrelated representation space, proved to be efficient in capturing the dynamics of typical proprioceptive control environments, in addition to being more computationally efficient through dimensionality reduction.

To derive practical applications of our findings, we tackled two RL use-cases: data-augmented off-policy RL, where our algorithm DICL-SAC improves the sample efficiency of SAC, and benefits from a theoretical guarantee under the framework of model-based multi-branch rollouts. Our second application, consisted in predicting the trajectory of rewards in order to perform hybrid online and model-based policy evaluation. Finally, we showed that the LLM-based dynamics model also provides well-calibrated uncertainty estimates.

## ACKNOWLEDGEMENTS

The authors extend their gratitude to Nicolas Boullé for insightful discussions on the initial concepts of this project, as well as to the authors of the paper (Liu et al., 2024c) (Toni J.B. Liu, Nicolas Boullé, Raphaël Sarfati, Christopher J. Earls) for providing access to their codebase. The authors also appreciate the anonymous reviewers and meta-reviewers for their valuable time and constructive feedback. This work was made possible thanks to open-source software, including Python (Van Rossum & Drake Jr, 1995), PyTorch (Paszke et al., 2019), Scikit-learn (Pedregosa et al., 2011), and CleanRL (Huang et al., 2022).

## REPRODUCIBILITY STATEMENT

In order to ensure reproducibility we release the code at https://github.com/abenechehab/dicl. The implementation details and hyperparameters are listed in Appendix D.

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

# Appendix

**Outline.** In Appendix A, we prove our main theoretical result (Theorem 4.2). We provide an extended related work in Appendix B. Additional materials about the state and action dimensions interdependence are given in Appendix C. The implementation details and hyperparameters of our methods are given in Appendix D. Finally, we provide additional experiments about multi-step errors (Appendix F), calibration (Appendix G), the impact of the data collecting policy on the prediction error (Appendix E), and details about the ablation study on the choice of the LLM (Appendix H).

## TABLE OF CONTENTS

# A   THEORETICAL ANALYSIS

## A.1   PROOF OF THEOREM 4.2

We start by formally defining the LLM multi-branch return $\eta_{p,k,T}^{\text{llm}}$. To do so, we first denote $A_t$ the random event of starting a $k$-step LLM branch at timestep $t$ and we denote $X_t$ the associated indicator random variable $X_t = \mathbb{1}[A_t]$. We assume that the $(X_t)_{t \geq T}$ are independent. We then define the random event $A_t^k$ that at least one of the $k$ preceding timesteps has been branched, meaning that the given timestep $t$ belongs to at least one LLM branch among the $k$ possible branches: $A_t^k = \bigcup_{i=0}^{k-1} A_{t-i}$. The LLM multi-branch return can then be written as follows:

$$
\eta_{p,k,T}^{\text{llm}}(\pi) = \underbrace{\sum_{t=0}^{T-1} \gamma^t \mathbb{E}_{s_t \sim P^t, a_t \sim \pi}\left[r(s_t, a_t)\right]}_{\text{Burn-in phase to gather minimal context size } T}
$$

$$
+ \sum_{t=T}^{\infty} \gamma^t \mathbb{E}_{X_{t-i} \sim b(p), 1 \leq i \leq k}\left[ \mathbb{1}[A_t^k] \underbrace{\frac{1}{\sum_{i=1}^{k} X_{t-i}} \sum_{i=1}^{k} X_{t-i} \mathbb{E}_{s_t \sim \hat{P}_{t,\text{llm}}^i, a_t \sim \pi}\left[r(s_t, a_t)\right]}_{\text{average reward among the branches spanning timestep } t} \right.
\tag{5}
$$

$$
+ \left. \underbrace{\mathbb{1}[\bar{A}_t^k] \mathbb{E}_{s_t \sim P^t, a_t \sim \pi}\left[r(s_t, a_t)\right]}_{\text{When no branch is spanning timestep } t} \right],
$$

where $P^t = P(.|P^{t-1})$ with $P^0 = \mu_0$ the initial state distribution and $\hat{P}_{t,\text{llm}}^i = \hat{P}_{\text{llm}}^i(.|P^{t-i})$.

Before continuing, we first need to establish the following lemma.

**Lemma A.1.** *(Multi-step Error Bound, Lemma B.2 in Frauenknecht et al. (2024) and Janner et al. (2019).) Let $P$ and $\tilde{P}$ be two transition functions. Define the multi-step error at time step $t$, starting from any initial state distribution $\mu_0$, as:*

$$
\varepsilon_t := D_{TV}(P^t(\cdot|\mu_0) \| \tilde{P}^t(\cdot|\mu_0))
$$

*with $P^0 = \tilde{P}^0 = \mu_0$.*
*Let the one-step error at time step $t \geq 1$ be defined as:*

$$
\xi_t := \mathbb{E}_{s \sim P^{t-1}(\cdot|\mu_0)}\left[D_{TV}(P(\cdot|s) \| \tilde{P}(\cdot|s))\right],
$$

*and $\xi_0 = \varepsilon_0 = 0$.*

*Then, the multi-step error satisfies the following bound:*

$$
\varepsilon_t \leq \sum_{i=0}^{t} \xi_i.
$$

*Proof.* Let $t > 0$. We start with the definition of the total variation distance:

$$
\begin{aligned}
\varepsilon_t &= D_{\text{TV}}(P^t(\cdot|\mu_0) \| \tilde{P}^t(\cdot|\mu_0)) \\
&= \frac{1}{2} \int_{s' \in \mathcal{S}} \left| P^t(s'|\mu_0) - \tilde{P}^t(s'|\mu_0) \right| ds' \\
&= \frac{1}{2} \int_{s' \in \mathcal{S}} \left| \int_{s \in \mathcal{S}} P(s'|s) P^{t-1}(s|\mu_0) - \tilde{P}(s'|s) \tilde{P}^{t-1}(s|\mu_0) \, ds \right| ds' \\
&\leq \frac{1}{2} \int_{s' \in \mathcal{S}} \int_{s \in \mathcal{S}} \left| P(s'|s) P^{t-1}(s|\mu_0) - \tilde{P}(s'|s) \tilde{P}^{t-1}(s|\mu_0) \right| ds \, ds' \\
&= \frac{1}{2} \int_{s' \in \mathcal{S}} \int_{s \in \mathcal{S}} \left| P(s'|s) P^{t-1}(s|\mu_0) - \tilde{P}(s'|s) \tilde{P}^{t-1}(s|\mu_0) \right| ds \, ds'
\end{aligned}
$$

$$\begin{aligned}
&= \frac{1}{2} \int_{s' \in \mathcal{S}} \int_{s \in \mathcal{S}} \Big| P(s'|s) P^{t-1}(s|\mu_0) - \tilde{P}(s'|s) P^{t-1}(s|\mu_0) \\
&\qquad + \tilde{P}(s'|s) P^{t-1}(s|\mu_0) - \tilde{P}(s'|s) \tilde{P}^{t-1}(s|\mu_0) \Big| \, ds \, ds' \\
&\leq \frac{1}{2} \int_{s' \in \mathcal{S}} \int_{s \in \mathcal{S}} P^{t-1}(s|\mu_0) \left| P(s'|s) - \tilde{P}(s'|s) \right| ds \, ds' \\
&\qquad + \frac{1}{2} \int_{s' \in \mathcal{S}} \int_{s \in \mathcal{S}} \tilde{P}(s'|s) \left| P^{t-1}(s|\mu_0) - \tilde{P}^{t-1}(s|\mu_0) \right| ds \, ds' \\
&= \int_{s \in \mathcal{S}} \left[ \frac{1}{2} \int_{s' \in \mathcal{S}} \left| P(s'|s) - \tilde{P}(s'|s) \right| ds' \right] P^{t-1}(s|\mu_0) \, ds \\
&\qquad + \frac{1}{2} \int_{s \in \mathcal{S}} \left( \int_{s' \in \mathcal{S}} \tilde{P}(s'|s) \, ds' \right) \left| P^{t-1}(s|\mu_0) - \tilde{P}^{t-1}(s|\mu_0) \right| ds \\
&= \mathbb{E}_{s \sim P^{t-1}(\cdot|\mu_0)} \left[ D_{\mathrm{TV}}(P(\cdot|\mu_0) \| \tilde{P}(\cdot|s)) \right] + D_{\mathrm{TV}}(P^{t-1}(\cdot|\mu_0) \| \tilde{P}^{t-1}(\cdot|\mu_0)) \\
&= \xi_t + \varepsilon_{t-1}
\end{aligned}$$

Given that $\xi_0 = \varepsilon_0 = 0$, by induction we have:

$$\varepsilon_t \leq \sum_{i=0}^{t} \xi_i.$$

$\square$

We now restate and prove Theorem 4.2:

**Theorem A.2** (Multi-branch return bound). *Let $T$ be the minimal length of the in-context trajectories, $p \in [0,1]$ the probability that a given state is a branching point. We assume that the reward is bounded and that the expected total variation between the LLM-based model and the true dynamics under a policy $\pi$ is bounded at each timestep by $\max_{t \geq T} \mathbb{E}_{s \sim P^t, a \sim \pi}[D_{TV}(P(.|s,a) \| \hat{P}_{llm}(.|s,a))] \leq \varepsilon_{llm}(T)$. Then under a multi-branched rollout scheme with a branch length of $k$, the return is bounded as follows:*

$$|\eta(\pi) - \eta_{p,k,T}^{llm}(\pi)| \leq 2 \frac{\gamma^T}{1-\gamma} r_{\max} k^2 \, p \, \varepsilon_{llm}(T) \ , \tag{6}$$

*where $r_{\max} = \max_{s \in \mathcal{S}, a \in \mathcal{A}} r(s,a)$.*

*Proof.* **Step 1: Expressing the bound in terms of horizon-dependent errors.**

$$\begin{aligned}
|\eta(\pi) - \eta_{p,k,T}^{\mathrm{llm}}(\pi)| &= \Bigg| \sum_{t=T}^{\infty} \gamma^t \mathbb{E}_{s_t \sim P^t, a_t \sim \pi} \big[ r(s_t, a_t) \big] \\
&\qquad - \mathbb{E}_{X_{t-i} \sim b(p), 1 \leq i \leq k} \bigg[ \mathbb{1}[A_t^k] \frac{1}{\sum_{i=1}^{k} X_{t-i}} \sum_{i=1}^{k} X_{t-i} \mathbb{E}_{s_t \sim \hat{P}_{t,\mathrm{llm}}^i, a_t \sim \pi} \big[ r(s_t, a_t) \big] \\
&\qquad - \mathbb{1}[\bar{A}_t^k] \mathbb{E}_{s_t \sim P^t, a_t \sim \pi} \big[ r(s_t, a_t) \big] \bigg] \Bigg| \\
&\leq \sum_{t=T}^{\infty} \gamma^t \Bigg| \mathbb{E}_{X_{t-i} \sim b(p), 1 \leq i \leq k} \bigg[ \mathbb{1}[A_t^k] \mathbb{E}_{s_t \sim P^t, a_t \sim \pi} \big[ r(s_t, a_t) \big] + \mathbb{1}[\bar{A}_t^k] \mathbb{E}_{s_t \sim P^t, a_t \sim \pi} \big[ r(s_t, a_t) \big] \bigg] \\
&\qquad - \mathbb{E}_{X_{t-i} \sim b(p), 1 \leq i \leq k} \bigg[ \mathbb{1}[A_t^k] \frac{1}{\sum_{i=1}^{k} X_{t-i}} \sum_{i=1}^{k} X_{t-i} \mathbb{E}_{s_t \sim \hat{P}_{t,\mathrm{llm}}^i, a_t \sim \pi} \big[ r(s_t, a_t) \big] \\
&\qquad - \mathbb{1}[\bar{A}_t^k] \mathbb{E}_{s_t \sim P^t, a_t \sim \pi} \big[ r(s_t, a_t) \big] \bigg] \Bigg|
\end{aligned}$$

$$\leq \sum_{t=T}^{\infty} \gamma^t \left| \mathbb{E}_{X_{t-i} \sim b(p), 1 \leq i \leq k} \left[ \mathbb{1}[A_t^k] \Big( \right. \right.$$

$$\left. \left. \mathbb{E}_{s_t \sim P^t, a_t \sim \pi} [r(s_t, a_t)] - \frac{1}{\sum_{i=1}^k X_{t-i}} \sum_{i=1}^k X_{t-i} \mathbb{E}_{s_t \sim \hat{P}_{t,\text{llm}}^i, a_t \sim \pi} [r(s_t, a_t)] \Big) \right] \right|$$

$$\leq \sum_{t=T}^{\infty} \gamma^t \left| \mathbb{E}_{X_{t-i} \sim b(p), 1 \leq i \leq k} \left[ \mathbb{1}[A_t^k] \frac{1}{\sum_{i=1}^k X_{t-i}} \sum_{i=1}^k X_{t-i} \Big( \right. \right.$$

$$\left. \left. \mathbb{E}_{s_t \sim P^t, a_t \sim \pi} [r(s_t, a_t)] - \mathbb{E}_{s_t \sim \hat{P}_{t,\text{llm}}^i, a_t \sim \pi} [r(s_t, a_t)] \Big) \right] \right|$$

We then expand the integrals in the terms $\mathbb{E}_{s_t \sim P^t, a_t \sim \pi} [r(s_t, a_t)] - \mathbb{E}_{s_t \sim \hat{P}_{t,\text{llm}}^i, a_t \sim \pi} [r(s_t, a_t)]$ and express it in terms of horizon-dependent multi-step model errors:

$$\mathbb{E}_{s_t \sim P^t, a_t \sim \pi} [r(s_t, a_t)] - \mathbb{E}_{s_t \sim \hat{P}_{t,\text{llm}}^i, a_t \sim \pi} [r(s_t, a_t)]$$

$$= \int_{s \in \mathcal{S}} \int_{a \in \mathcal{A}} r(s, a) \big( P^t(s, a) - \hat{P}_{t,\text{llm}}^i(s, a) \big) \, da \, ds$$

$$\leq r_{\max} \int_{s \in \mathcal{S}} \int_{a \in \mathcal{A}} \big( P^t(s, a) - \hat{P}_{t,\text{llm}}^i(s, a) \big) \, da \, ds$$

$$\leq r_{\max} \int_{s \in \mathcal{S}} \int_{a \in \mathcal{A}} \big( P^t(s) - \hat{P}_{t,\text{llm}}^i(s) \big) \pi(a|s) \, da \, ds \tag{7}$$

$$\leq r_{\max} \int_{s \in \mathcal{S}} \big( P^t(s) - \hat{P}_{t,\text{llm}}^i(s) \big) \, ds$$

$$\leq 2 r_{\max} D_{\text{TV}}(P^t || \hat{P}_{t,\text{llm}}^i)$$

**Step 2: Simplifying the bound.**
By applying Lemma A.1 we can bound the multi-step errors using the bound on one-step errors:

$$D_{\text{TV}}(P^t || \hat{P}_{t,\text{llm}}^i) \leq i \, \varepsilon_{\text{llm}}(T) \leq k \, \varepsilon_{\text{llm}}(T) \tag{8}$$

Therefore, the bound becomes:

$$|\eta(\pi) - \eta_{p,k,T}^{\text{llm}}(\pi)| \leq 2 r_{\max} k \, \varepsilon_{\text{llm}}(T) \sum_{t=T}^{\infty} \gamma^t \left| \mathbb{E}_{X_{t-i} \sim b(p), 1 \leq i \leq k} \left[ \mathbb{1}[A_t^k] \frac{1}{\sum_{i=1}^k X_{t-i}} \sum_{i=1}^k X_{t-i} \right] \right|$$

$$= 2 r_{\max} k \, \varepsilon_{\text{llm}}(T) \sum_{t=T}^{\infty} \gamma^t \left| \mathbb{E}_{X_{t-i} \sim b(p), 1 \leq i \leq k} \left[ \mathbb{1}[A_t^k] \right] \right|$$

$$\leq 2 r_{\max} k \, \varepsilon_{\text{llm}}(T) \sum_{t=T}^{\infty} \gamma^t k p$$

$$= 2 \frac{\gamma^T}{1 - \gamma} r_{\max} k^2 \, p \, \varepsilon_{\text{llm}}(T)$$

$$\tag{9}$$

$\square$

## B   RELATED WORK

**Model-based reinforcement learning (MBRL).**   MBRL has been effectively used in iterated batch RL by alternating between model learning and planning (Deisenroth & Rasmussen, 2011; Hafner et al., 2021; Gal et al., 2016; Levine & Koltun, 2013; Chua et al., 2018; Janner et al., 2019;

Kégl et al., 2021), and in the offline (pure batch) RL where we do one step of model learning followed by policy learning (Yu et al., 2020; Kidambi et al., 2020; Lee et al., 2021; Argenson & Dulac-Arnold, 2021; Zhan et al., 2021; Yu et al., 2021; Liu et al., 2021; Benechehab et al., 2024). Planning is used either at decision time via model-predictive control (MPC) (Draeger et al., 1995; Chua et al., 2018; Hafner et al., 2019; Pinneri et al., 2020; Kégl et al., 2021), or in the background where a model-free agent is learned on imagined model rollouts (Dyna; Janner et al. (2019); Sutton (1991); Sutton et al. (1992); Ha & Schmidhuber (2018)), or both. For example, model-based policy optimization (MBPO) (Janner et al., 2019) trains an ensemble of feed-forward models and generates imaginary rollouts to train a soft actor-critic agent.

**LLMs in RL.** LLMs have been integrated into reinforcement learning (RL) (Cao et al., 2024; Yang et al., 2023), playing key roles in enhancing decision-making (Kannan et al., 2024; Pignatelli et al.; Zhang et al., 2024; Feng et al., 2024), reward design (Kwon et al., 2023; Wu et al., 2024; Carta et al., 2023; Liu et al., 2023), and information processing (Poudel et al., 2023; Lin et al., 2024). The use of LLMs as world models is particularly relevant to our work. More generally, the Transformer architecture (Vaswani et al., 2017) has been used in offline RL (Decision Transformer Chen et al. (2021); Trajectory Transformer Janner et al. (2021)). Pre-trained LLMs have been used to initialize decision transformers and fine-tune them for offline RL tasks (Shi et al., 2023; Reid et al., 2022; Yang & Xu, 2024). As world models, Dreamer-like architectures based on Transformers have been proposed (Micheli et al., 2022; Zhang et al., 2023a; Chen et al., 2022), demonstrating efficiency for long-memory tasks such as Atari games. In text-based environments, LLMs have found multiple applications (Lin et al., 2024; Feng et al., 2024; Zhang et al., 2024; Ma et al., 2024), including using code-generating LLMs to generate policies in a zero-shot fashion (Liang et al., 2023; Liu et al., 2024a).

The closest work to ours is Wang et al. (2023), where a system prompt consisting of multiple pieces of information about the control environment (e.g., description of the state and action spaces, nature of the controller, historical observations, and actions) is fed to the LLM. Unlike our approach, which focuses on predicting the dynamics of RL environments, Wang et al. (2023) aim to directly learn a low-level control policy from the LLM, incorporating extra information in the prompt. Furthermore, Wang et al. (2023) found that only GPT-4 was usable within their framework, while we provide a proof-of-concept using smaller open LLMs such as Llama 3.2 1B.

**ICL on Numerical Data.** In-context learning for regression tasks has been theoretically analyzed in several works, providing insights based on the Transformer architecture (Li et al., 2023; von Oswald et al., 2023; Akyürek et al., 2023; Garg et al., 2023; Xie et al., 2022). Regarding time series forecasting, LLMTime (Gruver et al., 2023a) successfully leverages ICL for zero-shot extrapolation of one-dimensional time series data. Similarly, Das et al. (2024) introduce a foundational model for one-dimensional zero-shot time series forecasting, while Xue & Salim (2023) combine numerical data and text in a question-answer format. ICL can also be used to approximate a continuous density from the LLM logits. For example, Liu et al. (2024c) develop a Hierarchical *softmax* algorithm to infer the transition rules of uni-dimensional Markovian dynamical systems. Building on this work, Zekri et al. provide an application that predicts the parameter value trajectories in the Stochastic Gradient Descent algorithm. More relevant to our work, Requeima et al. (2024) presented *LLMProcesses*, a method aimed at extracting multi-dimensional distributions from LLMs. Other practical applications of ICL on numerical data include few-shot classification on tabular data (Hegselmann et al., 2023), regression (Vacareanu et al., 2024), and meta ICL (Coda-Forno et al., 2023).

# C STATE AND ACTION DIMENSIONS INTERDEPENDENCE - ADDITIONAL MATERIALS

## C.1 PRINCIPAL COMPONENT ANALYSIS (PCA)

**Principal Component Analysis.** PCA is a dimensionality reduction technique that transforms the original variables into a new set of variables, the principal components, which are linearly uncorrelated. The principal components can be ordered such that the first few retain most of the variation present in all of the original variables. Formally, given a data matrix $\mathbf{X}$ with $n$ observations and $p$ variables, PCA diagonalizes the covariance matrix $\mathbf{C} = \frac{1}{n-1}\mathbf{X}^T\mathbf{X}$ to find

the eigenvectors, which represent the directions of the principal components: PCA: $\mathbf{X} \rightarrow \mathbf{Z} = \mathbf{XW}$, where $\mathbf{W}$ are the eigenvectors of $\mathbf{C}$. In our case, the data represents a dataset of states and actions given a data collecting policy $\pi_D$, while the $p$ variables represent the state (eventually also the action) dimensions.

**Ablation on the number of components.** Fig. 9 shows an ablation study on the number of components used in the DICL-$(s, a)$ method. Surprisingly, we observe a sharp decline in the average multi-step error (see Appendix F for a detailed definition) given only 4 components among 23 in the HalfCheetah system. The error then slightly increases for an intermediate number of components, before going down again when the full variance is recovered. This finding strengthens the position of PCA as our Disentangling algorithm of choice in DICL.

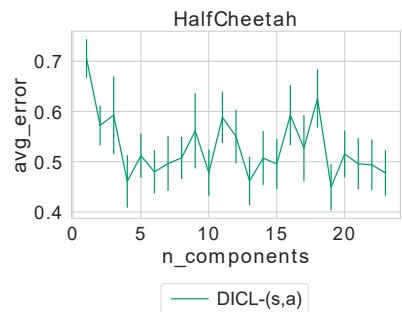

### C.2 INDEPENDENT COMPONENT ANALYSIS (ICA)

ICA is a statistical and computational technique used to separate a multivariate signal into additive, statistically independent components. Unlike PCA, which decorrelates the data, ICA aims to find a linear transformation

Figure 9: Ablation study on the number of principal components in the DICL-$(s, a)$ method.

that makes the components as independent as possible. Given a data matrix $\mathbf{X}$, ICA assumes that the data is generated as linear mixtures of independent components: $\mathbf{X} = \mathbf{AS}$, where $\mathbf{A}$ is an unknown mixing matrix and $\mathbf{S}$ is the matrix of independent components with independent rows. The goal of ICA is to estimate an unmixing matrix $\mathbf{W}$ such that $\mathbf{Y} = \mathbf{WX}$ is a good approximation of the independent components $\mathbf{S}$. The implications of ICA on independence are profound: while PCA only guarantees uncorrelated components, ICA goes a step further by optimizing for statistical independence, often measured by non-Gaussianity (kurtosis or negentropy).

Fig. 10 shows the estimated mixing matrix $\mathbf{A}$ when running ICA on the D4RL-*expert* dataset on the Hopper environment. Under the assumptions of ICA, notably the statistical independence of the source signals, their linear mixing and the invertibility of the original (unknown) mixing matrix, the original sources are successfully recovered if each line of the estimated mixing matrix is mostly dominated by a single value, meaning that it's close to an identity matrix up to a permutation with scaling. In the case of our states and actions data, it's not clear that this is the case from Fig. 10. Similarly to PCA, we can transform the in-context multi-dimensional signal using ICA, and apply the ICL procedure to the recovered independent sources. We plan on exploring this method in future follow-up work.

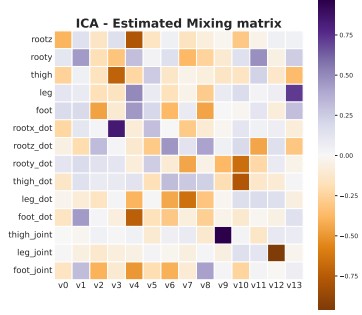

Figure 10: ICA estimated mixing matrix.

### C.3 AUTOENCODER-BASED APPROACH

Variational Autoencoders (VAEs) (Kingma & Welling, 2022) offer a powerful framework for learning representations. A disentangled representation is one where each dimension of the latent space captures a distinct and interpretable factor of variation in the data. By combining an encoder network that maps inputs to a probabilistic latent space with a decoder network that reconstructs the data, VAEs employ the reparameterization trick to enable backpropagation through the sampling process. The key to disentanglement lies in the KL-divergence term of the VAE loss function, which regularizes the latent distribution to be close to a standard normal distribution. Variants such as $\beta$-VAE (Higgins et al., 2017) further emphasize this regularization by scaling the KL-divergence term, thereby encouraging the model to learn a more disentangled representation at the potential cost of reconstruction quality. Beyond simple VAEs, there exist previous work in the literature that specifically aim at learning a factorized posterior distribution in the latent space (Kim & Mnih, 2019).

Although this direction looks promising, it strikes different concerns about the learnability of these models in the low data regime considered in our paper.

## C.4 SENSITIVITY ANALYSIS

The preceding analysis examines state dimensions as features within a representation space, disregarding their temporal nature and our ultimate objective of predicting the next state. In practice, our interest lies in capturing the dependencies that most significantly influence the next state through the dynamics function of the MDP. To achieve this, we use Sensitivity Analysis (SA) to investigate how variations in the input of the dynamics function impact its output.

**Sensitivity Analysis.** Sensitivity analysis is a systematic approach to evaluate how the uncertainty in the output of a model can be attributed to different sources of uncertainty in the model's inputs. The One-at-a-Time (OAT) method is a technique used to understand the impact of individual input variables on the output of a model. In the context of a transition function of a MDP, the OAT method involves systematically varying one current state or action dimension at a time, while keeping all others fixed, and observing the resulting changes in the output dimensions: $\frac{\partial (\mathbf{s}_{t+1})_k}{\partial (\mathbf{s}_t)_i}$ and $\frac{\partial (\mathbf{s}_{t+1})_k}{\partial (\mathbf{a}_t)_j}$, where $(\mathbf{s}_t)_i$, $(\mathbf{a}_t)_j$ and $(\mathbf{s}_{t+1})_k$ denote the $i$-th dimension of the state, the $j$-th dimension of the action, and the $k$-th dimension of the next state, respectively.

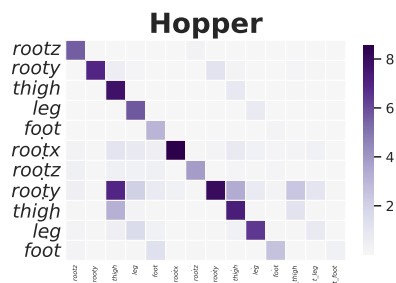

Figure 11: Sensitivity matrix.

In practice, we measure the sensitivity by applying a perturbation (of scale $10\%$) to each input dimension separately, reporting the absolute change that occurs in each dimension of the output. Precisely, for a deterministic transition function $f$, input state dimension $i$, and output dimension $k$, we measure $|f(s + \epsilon, a)_k - f(s, a)_k|$ where $\epsilon_i = 0.1 \times scale(i)$ and 0 elsewhere. The sensitivity matrix in Fig. 11 demonstrates that most of the next state dimensions are mostly affected by their respective previous values (the diagonal shape in the state dimensions square). In addition to that, actions only directly affect some state dimensions, specifically velocities, which is expected from the nature of the physics simulation underlying those systems. This finding suggests that the vICL method might give good results in practice for the considered RL environments, and makes us hope that the DICL-$(s)$ approach is enough to capture the state dimensions dependencies, especially for single-step prediction.

*Remark* C.1. This sensitivity analysis is specific to the single-step transition function. In practice, such conclusions might change when looking at a larger time scale of the simulation.

## D ALGORITHMS

### D.1 SOFT-ACTOR CRITIC

Soft Actor-Critic (SAC) (Haarnoja et al., 2018) is an off-policy algorithm that incorporates the maximum entropy framework, which encourages exploration by seeking to maximize the entropy of the policy in addition to the expected return. SAC uses a deep neural network to approximate the policy (actor) and the value functions (critics), employing two Q-value functions to mitigate positive bias in the policy improvement step typical of off-policy algorithms. This approach helps in learning more stable and effective policies for complex environments, making SAC particularly suitable for tasks with high-dimensional, continuous action spaces.

We use the implementation provided in CleanRL (Huang et al., 2022) for SAC. In all environments, we keep the default hyperparameters provided with the library, except for the update frequency. We specify in Table 2 the complete list of hyperparameters used for every considered environment.

Table 2: SAC hyperparameters.

| Environment | HalfCheetah | Hopper | Pendulum |
|---|---|---|---|
| Update frequency | 1000 | 1000 | 200 |
| Learning starts | 5000 | 5000 | 1000 |
| Batch size | 128 | 128 | 64 |
| Total timesteps | $1e6$ | $1e6$ | $1e4$ |
| Gamma $\gamma$ | 0.99 | 0.99 | 0.99 |
| policy learning rate | $3e-4$ | $3e-4$ | $3e-4$ |

## D.2 DICL-SAC

For our algorithm, we integrate an LLM inference interface (typically the Transformers library from Huggingface (Wolf et al., 2020)) with CleanRL (Huang et al., 2022). Table 3 shows all DICL-SAC hyperparameter choices for the considered environments.

Table 3: DICL-SAC hyperparameters.

| Environment | HalfCheetah | Hopper | Pendulum |
|---|---|---|---|
| Update frequency | 1000 | 1000 | 200 |
| Learning starts | 5000 | 5000 | 1000 |
| LLM Learning starts | 10000 | 10000 | 2000 |
| LLM Learning frequency | 256 | 256 | 16 |
| Batch size | 128 | 128 | 64 |
| LLM Batch size ($\alpha\%$) | $7(5\%), 13(10\%), 32(25\%)$ | $7(5\%), 13(10\%), 32(25\%)$ | $4(5\%), 7(10\%), 16(25\%)$ |
| Total timesteps | $1e6$ | $1e6$ | $1e4$ |
| Gamma $\gamma$ | 0.99 | 0.99 | 0.99 |
| Max context length | 500 | 500 | 198 |
| Min context length | 1 | 1 | 1 |
| LLM sampling method | $mode$ | $mode$ | $mode$ |
| LLM dynamics learner | vICL | vICL | vICL |

**Balancing gradient updates.** To ensure that DICL-SAC performs equally important gradient updates on the LLM generated data, we used a gradient updates balancing mechanism. Indeed, since the default reduction method of loss functions is averaging, the batch $\mathcal{B}$ with the smallest batch size gets assigned a higher weight when doing gradient descent: $\frac{1}{|\mathcal{B}|}$. To address this, we multiply the loss corresponding to the LLM generated batch $\mathcal{B}_{\text{llm}}$ with a correcting coefficient $\frac{|\mathcal{B}_{\text{llm}}|}{|\mathcal{B}|}$ ensuring equal weighting across all samples.

We now show the full training curves on the HalfCheetah and Hopper environments (Fig. 12). The return curves show smoothed average training curves $\pm$ 95% Gaussian confidence intervals for 5 seeds in HalfCheetah and Hopper, and 10 seeds for Pendulum.

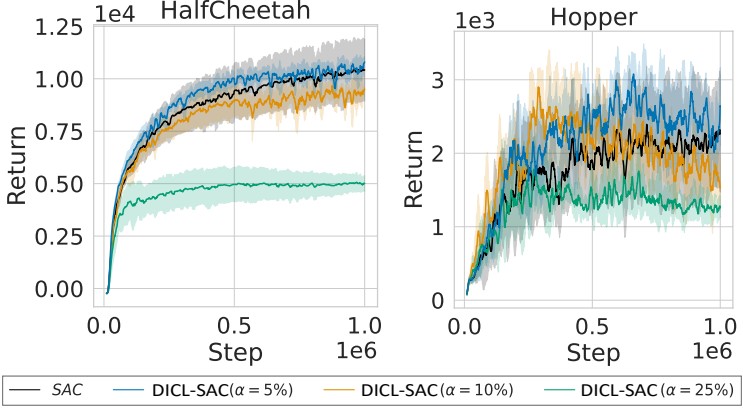

Figure 12: **Data-augmented off-policy RL.** Full training curves.

**The update frequency.** The default update frequency of SAC is 1 step, meaning that the policy that interacts with the environment gets updated after every interaction. In our LLM-based framework, this introduces an additional layer of complexity at this implies that the state visitation distribution of the in-context trajectories will be moving from one timestamp to another. We therefore assume an update frequency equal to the maximal number of steps of an episode of a given environment.

It is important to mention that the choice of setting the update frequency for all algorithms to the number of steps equivalent to a full episode has dual implications: it can stabilize the data collection policy, which is beneficial, but it may also lead to overtraining on data gathered by early, low-quality policies, which is detrimental. This trade-off has been previously studied in the RL literature (Matsushima et al., 2021; Thomas et al., 2024). Notably, Thomas et al. (2024) argues that the update frequency is more of a system constraint than a design choice or hyperparameter. For instance, controlling a physically grounded system, such as a helicopter, inherently imposes a minimal update frequency. Therefore, we deem it a fair comparison as this constraint is uniformly applied to all algorithms.

For the sake of completeness and comparison, we also evaluated the SAC baseline using its default update frequency of one step. Fig. 13 shows the comparison of our algorithm DICL-SAC, the baseline SAC with update frequency 1000, and the default SAC with update frequency 1. We see that on Halfcheetah the default SAC ($uf = 1$) performs similarly to SAC with an update frequency of 1000. On Pendulum and Hopper it performs slightly better with DICL remaining competitive while having the constraint of an update frequency of 1000.

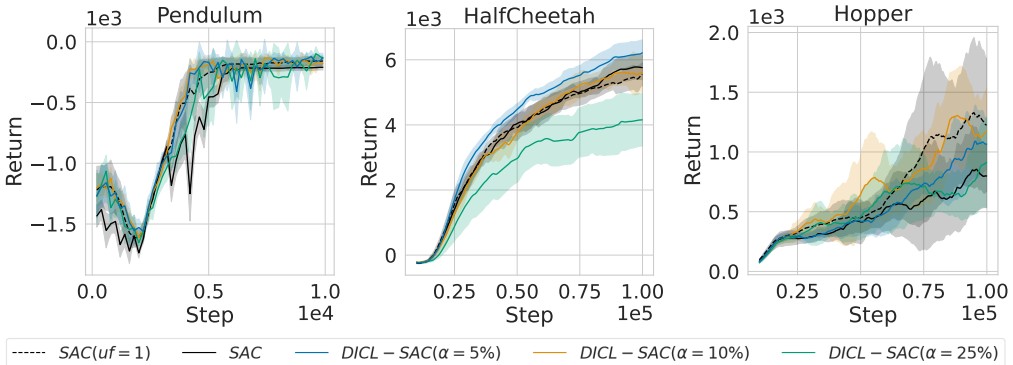

Figure 13: **Data-augmented off-policy RL.** Comparison with SAC in the default update frequency regime. We conducted this experiment using the *Llama 3.2-1B* model.

## E    WHAT IS THE IMPACT OF THE POLICY ON THE PREDICTION ERROR?

In this experiment, We investigate how a policy impacts the accuracy and calibration of our LLM-based dynamics models. To do so, we train three model-free algorithms (PPO (Schulman et al., 2017), SAC (Haarnoja et al., 2018), and TD3 (Fujimoto et al., 2019)) on the HalfCheetah environment, selecting different checkpoints throughout training to capture diverse policies. We then analyze the correlation between policy characteristics, specifically state coverage (defined as the maximum distance between any two states encountered by the policy) and entropy, with the Mean Squared Error and Kolmogorov-Smirnov (KS) statistic. Our findings indicate that the state coverage correlates with both MSE and KS, possibly because policies that explore a wide range of states generate trajectories that are more difficult to learn. Regarding the entropy, we can see that it also correlates with MSE, but interestingly, it does not appear to impact the calibration.

## F    MULTI-STEP PREDICTION ERRORS

**The average multi-step error.**    In Fig. 4a, we compute the average Mean Squared Error over prediction horizons for $h = 1, \ldots, 20$, and 5 trajectories sampled uniformly from the D4RL expert dataset. For visualization purposes, we first rescale all the dimensions (using a pipeline composed

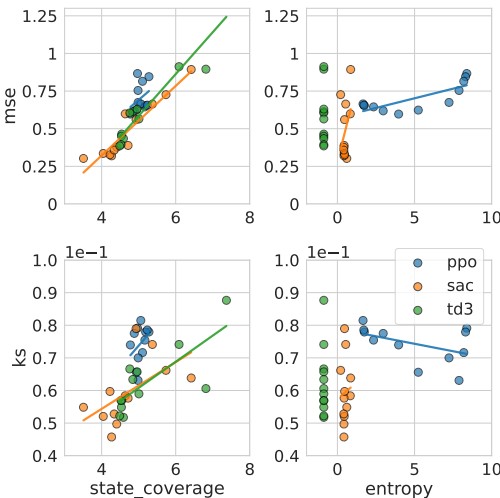

Figure 14: Correlation plots between state coverage and entropy of policies with MSE and KS metrics under the vICL dynamics learner.

of a *MinMaxScaler* and a *StandardScaler*) so that the respective MSEs are on the same scale. The MSE metric in Table 1 is also computed in a similar fashion, with the exception that it's average over 7 different tasks (HalfCheetah: random, medium, expert; Hopper: medium, expert; Walker2d: medium, expert).

**The MLP baseline.** For the $MLP$ baseline, we instantiate an MLP with: 4 layers, 128 neurons each, and $ReLU$ activations. We then format the in-context trajectory as a dataset of $\{(s_t, a_t, s_{t+1})\}$ on which we train the MLP for 150 epochs using early stopping and the Adam optimizer (Kingma & Ba, 2015).

We now extend Fig. 4 to show the multi-step generated trajectories for all the dimensions of the HalfCheetah system in Fig. 15.

# G  CALIBRATION

**The naive baseline.** In the calibration plots Figs. 8 and 16, we compare the LLM-based dynamics models with a (naive) baseline that estimates a Gaussian distribution using the in-context moments (mean and variance).

**KOLMOGOROV-SMIRNOV STATISTIC (KS)**: This metric is computed using the quantiles (under the model distribution) of the ground truth values. Hypothetically, these quantiles are uniform if the error in predicting the ground truth is a random variable distributed according to a Gaussian with the predicted standard deviation, a property we characterize as *calibration*. To assess this, we compute the Kolmogorov-Smirnov (KS) statistics. Formally, starting from the model cumulative distribution function (CDF) $F_\theta(s_{t+1}|s_t, a_t)$, we define the empirical CDF of the quantiles of ground truth values by $\mathcal{F}_{\theta,j}(x) = \frac{\left|\left\{(s_t, a_t, s_{t+1}) \in \mathcal{D} | F_\theta^j(s_{t+1}|s_t, a_t) \leq x\right\}\right|}{N}$ for $x \in [0, 1]$. We denote by $U(x)$ the CDF of the uniform distribution over the interval $[0, 1]$, and we define the KS statistics as the largest absolute difference between the two CDFs across the dataset $\mathcal{D}$:

$$\text{KS}(\mathcal{D}; \theta; j \in \{1, \ldots, d_s\}) =$$
$$\max_{i \in \{1, \ldots, N\}} \left| \mathcal{F}_{\theta,j}(F_\theta^j(s_{i,t+1}|s_{i,t}, a_{i,t})) - U(F_\theta^j(s_{i,t+1}|s_{i,t}, a_{i,t})) \right| \quad (10)$$

The KS score ranges between zero and one, with lower values indicating better calibration.

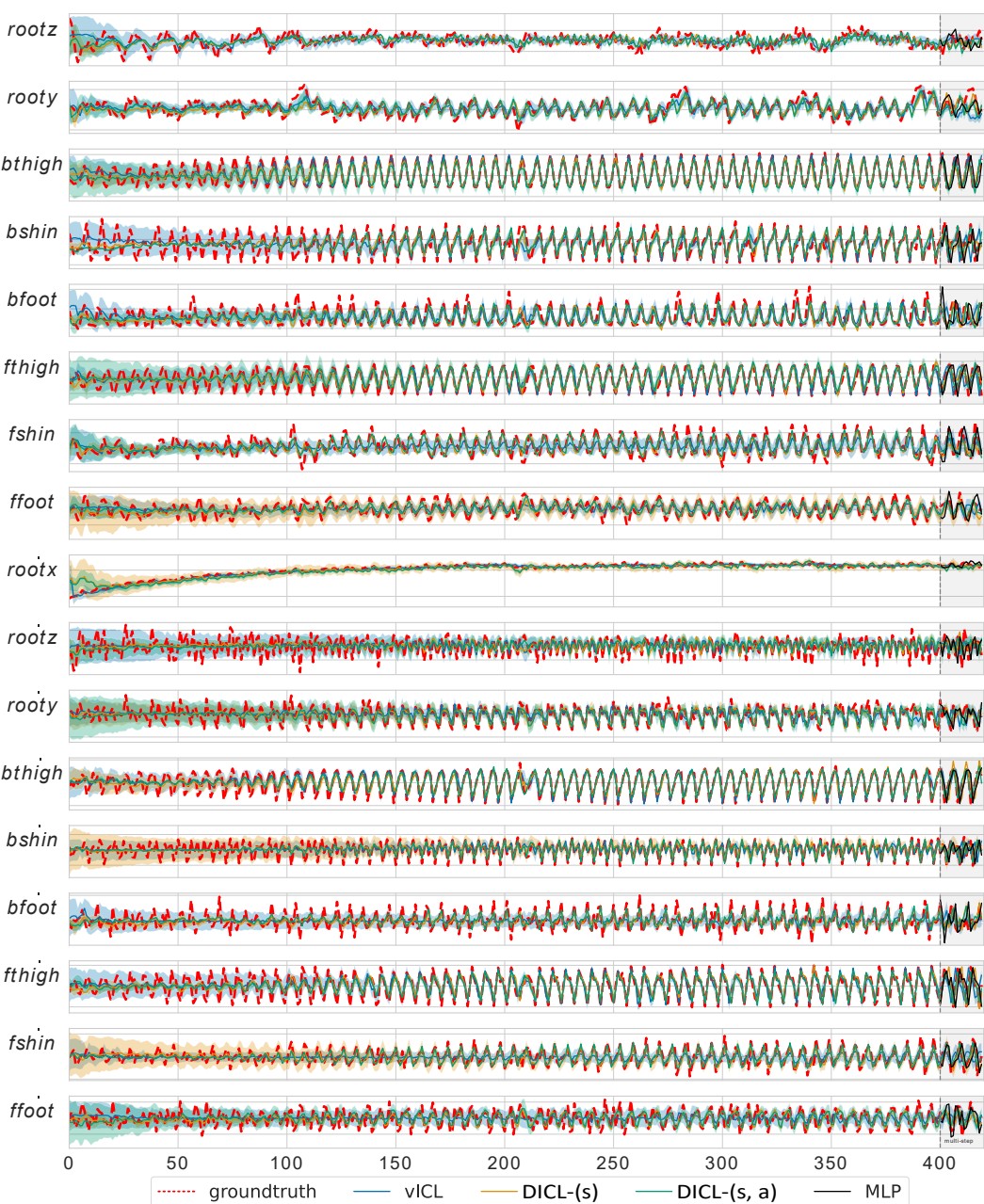

Figure 15: Halfcheetah

## H ON THE CHOICE OF THE LLM

In this ablation study, we investigate the impact of LLM size on prediction performance and calibration on D4RL tasks. The LLMs analyzed are all from the LLaMA 3 family of models (Dubey & the Llama 3 team., 2024), with size range from 1B to 70B parameters, including intermediate sizes of 3B and 8B. Each model is fed with 5 randomly sampled trajectories of length $T = 300$ from the D4RL datasets: expert, medium, and random. This latter task is only evaluated on HalfCheetah, since the Hopper and Walker2d environments random policies episodes do not have enough context yet to apply DICL. For the medium and expert datasets, we evaluate them on all the environments HalfCheetah (Fig. 17), Hopper (Fig. 18a), and Walker2d (Fig. 18b). The metrics used to evaluate the models are:

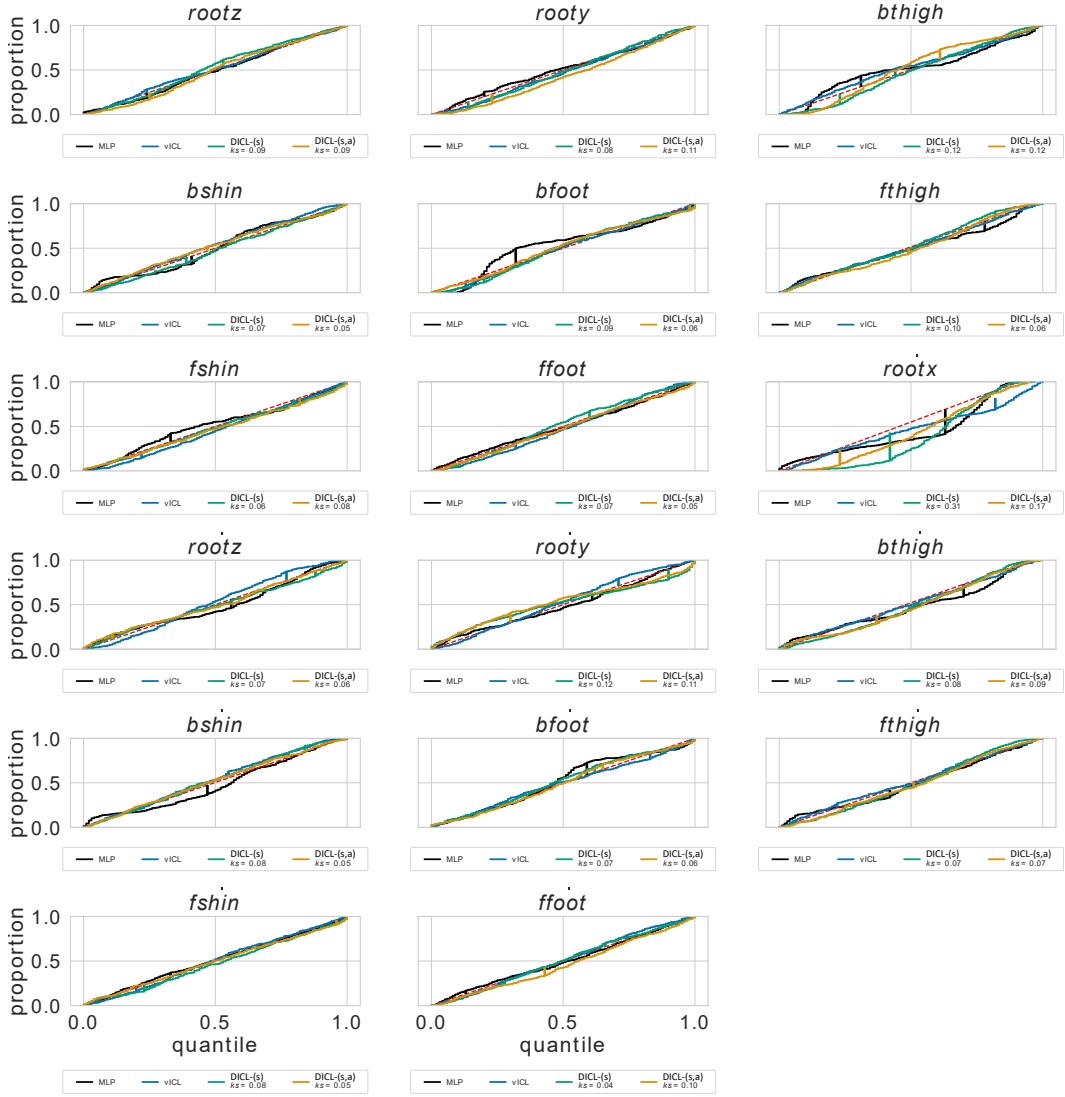

Figure 16: Halfcheetah

- Mean Squared Error (MSE): Applied after rescaling the data similarly to Appendix F to measure the prediction error.

- Kolmogorov-Smirnov (KS) statistic: To evaluate calibration, indicating how well the predicted probabilities match the observed outcomes. This metric is formally described in Appendix G.

All results are averaged over prediction horizons $h \in \{1, \dots, 20\}$. In the HalfCheetah environment, we observe that DICL-$(s)$ consistently outperforms the other variants across all tasks and with almost all LLMs in terms of prediction error. DICL-$(s, a)$ is outperformed by vICL in the *random* and *medium* datasets, while its performance improves in the *expert* dataset. This is likely because the policy has converged to a stable expert policy, making it easier for DICL-$(s, a)$ to predict actions as well. Regarding calibration, the three methods generally perform similarly, with a slight advantage for DICL-$(s, a)$, especially with smaller LLMs. In the Hopper environment, the MSE improvement of DICL over vICL is less pronounced with the smallest LLMs but becomes more evident with the LLaMA 3.1 70B model. However, DICL-$(s, a)$ consistently and significantly outperforms both vICL and DICL-$(s)$ in terms of the KS statistic (calibration). In the Walker2d environment, vICL proves to be a strong baseline in the expert task, while DICL-$(s)$ shows improvements over it in

the medium dataset. For calibration in Walker2d, DICL-$(s, a)$ continues to outperform the other variants across all tasks and LLM sizes.

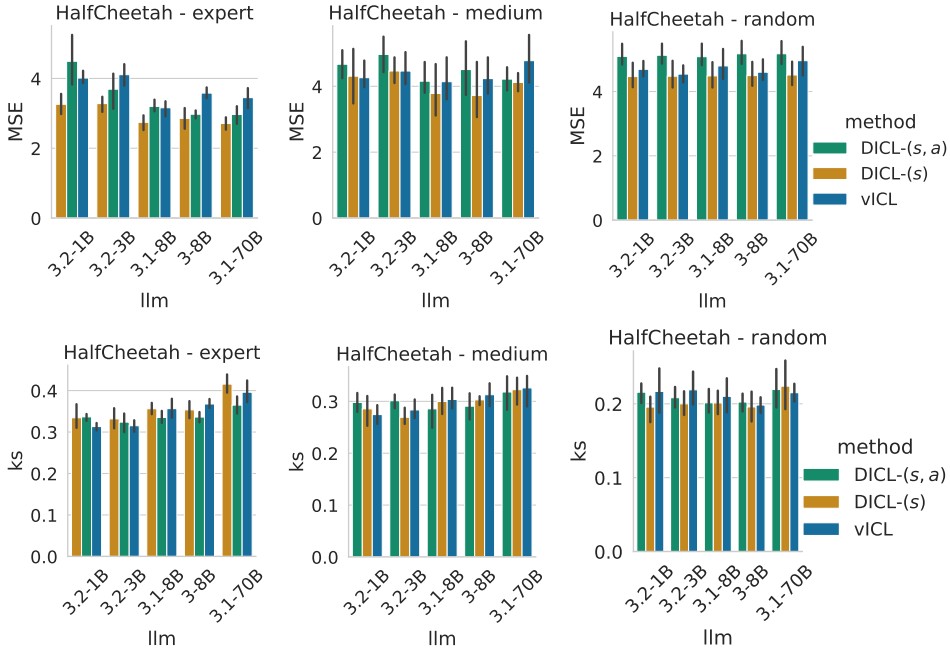

Figure 17: HalfCheetah.

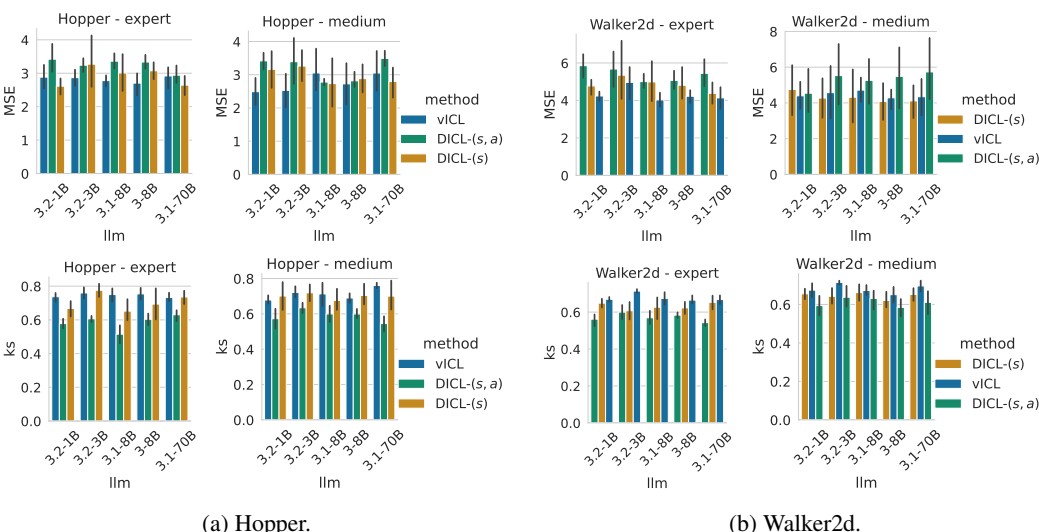

(a) Hopper.                    (b) Walker2d.

