# OpenReview forum: "Zero-shot Model-based Reinforcement Learning using Large Language Models"
_ICLR.cc/2025/Conference — ICLR 2025 Poster_

### Official Review · Reviewer_fXTe · 2024-10-30

**Soundness:** 3
**Presentation:** 3
**Contribution:** 3
**Rating:** 8
**Confidence:** 2

**Summary:**

The paper investigates zero-shot reinforcement learning (RL) for continuous control tasks using large language models (LLMs) and introduces "Disentangled In-Context Learning" (DICL) to improve LLMs' handling of continuous MDPs. This approach separates state and action features to better utilize in-context learning, with experiments validating its efficacy.

**Strengths:**

1. The authors proposed an innovative integration of LLMs into continuous reinforcement learning tasks.
2. Clear theoretical foundation supporting the DICL framework is shown.
3. The authors conduct extensive experiments to verify the effectiveness of the method across diverse RL scenarios.
4. The authors also did insightful analysis of zero-shot capabilities in complex MDPs.

**Weaknesses:**

Unclear experimental settings. The authors mentioned that they used 3 Llama3 models for the experiments, but all the experiment figures only show one result without stating which model is used. Also, an ablation study about the effect of different LLMs is missing.

**Questions:**

You mentioned that you change the update step of SAC to accommodate LLM's requirements. Is this unfair? Have you compared with a baseline SAC with update step 1?

---

> ### Author Response · Authors · 2024-11-19
> **Answer to Reviewer fXTe**
>
> We thank the reviewer fXTe for their positive and encouraging feedback. We are happy to read that the reviewer found our work innovative, well-supported by clear theoretical foundations and extensive experiments.
>
> We address the reviewer's concerns point by point below.
>
> >1. Unclear experimental settings. The authors mentioned that they used 3 Llama3 models for the experiments, but all the experiment figures only show one result without stating which model is used. Also, an ablation study about the effect of different LLMs is missing.
>
> We thank the Reviewer fXTe for their comment that enabled us to improve the clarity of our work. We updated the captions of each figure in the revised paper to make explicit the Llama version used in the experiments. Overall, we used the LLaMA 3-8B model for all experiments, except for DICL-SAC, which is LLM inference-intensive, and where we decided to use the more lightweight model LLaMA 3.2-1B.
>
> In addition, we thank the reviewer for pointing out the lack of an ablation study regarding the LLM choice in the DICL framework. To address this concern, we have included the Table 1 (p.6) in the revised paper that compares different version of LLaMA LLMs (3-8B, 3.1-8B, 3.1-70B, 3.2-1B, 3.2-3B) in terms of multi-step error (computed as the MSE of the forecasts at a prediction horizon of 20) and their calibration (computed as the Kolmogorov-Smirnov statistic). These metrics are averaged over 5 randomly selected episodes from each one of 7 D4RL [1] tasks (HalfCheetah: random, medium, expert; Hopper: medium, expert; Walker2d: medium, expert).
>
> We also included a discussion on this ablation study in Section 3.4, with more details deferred to Appendix H.
>
> >2. You mentioned that you change the update step of SAC to accommodate LLM's requirements. Is this unfair? Have you compared with a baseline SAC with update step 1?
>
> We opted to set the update frequency for all algorithms to the number of steps equivalent to a full episode (1000 steps instead of 1 step). This choice has dual implications: it can stabilize the data collection policy, which is beneficial, but it may also lead to overtraining on data gathered by early, low-quality policies, which is detrimental. This trade-off has been previously studied in the reinforcement learning literature [2,3]. Notably, [2] argues that the update frequency is more of a system constraint than a design choice or hyperparameter. For instance, controlling a physically grounded system, such as a helicopter, inherently imposes a minimal update frequency. Therefore, we deem it a fair comparison as this constraint is uniformly applied to all algorithms.
>
> We have added in Appendix D.2 a discussion and a figure that includes the default SAC configuration (with an update frequency of 1) to put it into perspective against DICL-SAC and the SAC baseline (both with an update frequency of 1000). We see that on Halfcheetah SAC with an update frequency of 1 performs similarly as SAC with an update frequency of 1000. On Pendulum and Hopper it performs slightly better with DICL remaining competitive while having the constraint of an update frequency of 1000.
>
> We hope that our comments addressed all the reviewer's concerns. We would be happy to provide additional comments upon the reviewer's request.
>
> ---
> - [1] Fu, Justin, et al. "D4rl: Datasets for deep data-driven reinforcement learning." arXiv preprint arXiv:2004.07219 (2020).
> - [2] A Thomas, A Benechehab, G Paolo, B Kégl. "Fair Model-Based Reinforcement Learning Comparisons with Explicit and Consistent Update Frequency" The Third Blogpost Track at ICLR 2024
> - [3] Matsushima, Tatsuya, et al. "Deployment-efficient reinforcement learning via model-based offline optimization." arXiv preprint arXiv:2006.03647 (2020).
> ---

---

> > ### Author Response · Authors · 2024-11-25
> > **Follow-up**
> >
> > We once again thank reviewer fXTe for their insightful feedback, which has helped to enhance the quality of the paper. We hope our reply has sufficiently addressed their comments. As the author-reviewer discussion period is ending, we would be glad to read their reply and address any additional questions.

---

### Official Review · Reviewer_D8FE · 2024-11-04

**Soundness:** 3
**Presentation:** 2
**Contribution:** 3
**Rating:** 6
**Confidence:** 3

**Summary:**

This paper aims to leverage large language models (LLMs) to achieve model-based reinforcement learning (MBRL). Specifically, it explores using the in-context learning (ICL) capability of LLMs to autoregressively predict the next state and reward, denoted as $<s_t, r_t>$, based on a given prior states and actions.

The authors' first approach, vanilla ICL (vICL), simplifies the objective by predicting $s_t$ solely based on prior states $s_{1:t-1}$, without considering the action. The second approach, DICL, uses Principal Component Analysis (PCA) to map the state-action vector to a latent space, where ICL is applied to the latent vectors. Additionally, they apply their LLM-based dynamics learner to augment the replay buffer of the Soft Actor-Critic (SAC) algorithm.

Another important contribution of this work is demonstrating the bound on the difference between true dynamics and LLM-based dynamics under a multi-branch rollout setting.

**Strengths:**

(please also see the summary)
1. This work systematically studies LLM-based MBRL.
2. The method is clearly presented and easy to follow.

**Weaknesses:**

The “zero-shot” claim is potentially misleading. If "zero-shot" is defined at the trajectory level, it is true that no trajectory-level examples were shown to the LLM during prediction. However, as shown in Section 4 (theoretical analysis), it appears necessary to use true dynamics to predict the transition and reward for steps $t < T$. These transitions, such as from $<s_{t-1}, a_{t-1}>$ to $s_t$, effectively serve as state-level few-shot examples. In my understanding, a true “zero-shot” setting would require that all previous transitions be predicted by the LLM itself autoregressively.

Another related concern is that the experimental setup is difficult to understand. For instance, Figure 3b was unclear to me. Given that the authors claim their method is zero-shot, I was unsure why, in the zero-shot setting, the agreement between the ground truth and the LLM-based method improves over time significantly. Wouldn’t error accumulation or distributional shift cause the LLM’s performance to degrade as more time steps are taken? After reviewing the theoretical analysis section, it seems likely that the result in Figure 3b is due to true dynamics being incorporated into the predictions, which was not mentioned. Please let me know if my understanding is inaccurate.

Finally, the ICL of LLM leverages the whole information in the previous context to predict the next transition. However, the MLP method can only accept the information of previous step as the input. The comparison is somehow unfair.

**Questions:**

n/a

---

> ### Author Response · Authors · 2024-11-19
> **Answer to Reviewer D8FE (Part 1/2)**
>
> We thank the reviewer D8FE for their insightful feedback. We are happy that the reviewer acknowledged that our study was systematic and clearly presented with an important theoretical contribution on the multi-branch rollouts bound.
>
> We address the reviewer's comments point by point below.
>
> >1. The “zero-shot” claim is potentially misleading. If "zero-shot" is defined at the trajectory level, it is true that no trajectory-level examples were shown to the LLM during prediction. However, as shown in Section 4 (theoretical analysis), it appears necessary to use true dynamics to predict the transition and reward for steps $t<T$. These transitions, such as from $(s_{t-1},a_{t-1})$ to $s_t$, effectively serve as state-level few-shot examples. In my understanding, a true “zero-shot” setting would require that all previous transitions be predicted by the LLM itself autoregressively.
>
> We appreciate Reviewer D8FE's feedback and acknowledge that the term 'zero-shot' within the ICL paradigm may not be entirely clear. Our use of "zero-shot" follows the literature on LLMs and time series, e.g. [1], where it refers to the fact that we perform no gradient updates or fine-tuning of the pretrained LLM's weights. The model operates purely through In-Context Learning (ICL), where we feed it a trajectory of length $T$ that serves as the "prompt" or query input, and ask it to predict the value at timestep $T+1$ as the model's zero-shot response. More precisely, in [2], the distinction is made between the *supervised learning* formulation of ICL --when the prompt is built using few supervised examples such as "$x_1 \to y_1, x_2 \to y_2, \ldots$"-- and the *dynamical systems* formulation of ICL -- where the query is the trajectory "$x_1, x_2, \ldots, x_T$" and the label is the next value $x_{T+1}$. In our approach, we endorse the latter formulation.
>
> To adress the reviewer's concern, we propose to add these explanations in the revised version of the paper to ensure clarity and accuracy in our presentation. Would such clarification make the use of the term "zero-shot" more intuitive for the reviewer?
>
> ---
> - [1] Gruver, Nate, et al. "Large language models are zero-shot time series forecasters." Advances in Neural Information Processing Systems 36 (2024).
> - [2] Li, Yingcong, et al. "Transformers as algorithms: Generalization and stability in in-context learning." International Conference on Machine Learning. PMLR, 2023.
> ---
>
> >2. Another related concern is that the experimental setup is difficult to understand. For instance, Figure 3b was unclear to me. Given that the authors claim their method is zero-shot, I was unsure why, in the zero-shot setting, the agreement between the ground truth and the LLM-based method improves over time significantly. Wouldn’t error accumulation or distributional shift cause the LLM’s performance to degrade as more time steps are taken? After reviewing the theoretical analysis section, it seems likely that the result in Figure 3b is due to true dynamics being incorporated into the predictions, which was not mentioned. Please let me know if my understanding is inaccurate.
>
> Regarding the experimental setup, Figure 3.b illustrates the quality of the forecast from DICL given a trajectory of length $t \in \{1,...,T\}$ (see timesteps 0 to 400, prior to the vertical line). This is equivalent to conducting an ablation study on the context length $T$ of the trajectory provided to the LLM as a query input. In more details, the LLM is simply called in parallel on all trajectories $x_1, x_2, ..., x_t$ for $t \in \{1,...,T\}$, and the output is compared against the corresponding next value $x_{t+1}$. As in time series, increasing the lookup window size improves the forecasting performance, hence the improvement over time, noticed by Reviewer D8FE.
>
> After the vertical red line at timestep 400, we perform autoregressive multi-step prediction (hence the label *multi-step* in the figure), where the LLM's predictions are concatenated to the in-context trajectory, and next values are predicted autoregressively. During this phase, we observe that by the forecast horizon of 20, the LLM predictions start deviating from the ground truth trajectories due to error accumulation, as mentioned by Reviewer D8FE.
>
> In accordance with the theoretical analysis, $T$ corresponds to the context length after which the LLM prediction error becomes small (in practice, $T \approx 300$ is sufficient). Therefore, in theory, we aim to restrict the LLM dynamics-based branches to states that occur after $T$ steps to ensure that the model error term $\varepsilon_\text{llm}(T)$ is small. In practice, however, we set $T=1$, considering all states as potential branching points.

---

> ### Author Response · Authors · 2024-11-19
> **Answer to Reviewer D8FE (Part 2/2)**
>
> >3. Finally, the ICL of LLM leverages the whole information in the previous context to predict the next transition. However, the MLP method can only accept the information of previous step as the input. The comparison is somehow unfair.
>
> We agree with Reviewer D8FE that at test time, the LLM is shown full trajectories while the MLP baseline is only shown the previous state and action. However, given the Markovian property of the studied systems, the full trajectory is not needed in theory, as the following holds:
>
> $P(s_t|a_{t-1},s_{t-1},...,s_1,a_0,s_0) = P(s_t|a_{t-1},s_{t-1})$
>
> Furthermore, the same in-context trajectory used for the LLM is also employed as training data for the MLP baseline. This ensures that both methods have access to the same data prior to evaluation, maintaining a fair comparison. Finally, it is important to mention that the MLP baseline natively and jointly models the non-linear dependencies between the state and action dimensions, unlike DICL, where we only capture their linear correlations through PCA. We hope our explanations can convince the reviewer of the validity of the comparison. In case it doesn't, does the reviewer have some suggestions to make the comparison more fair?
>
> We hope that all the reviewer's concerns and questions have been addressed and we remain open to future discussions. Given our explanations and revisions of the paper, we would be grateful if the reviewer could reconsider the evaluation of our work accordingly.

---

> ### Author Response · Authors · 2024-11-25
> **Follow-up**
>
> We once again thank reviewer D8FE for their insightful feedback, which has helped to enhance the quality of the paper. We hope our reply has sufficiently addressed their comments. As the author-reviewer discussion period is ending, we would be glad to read their reply and address any additional questions.

---

> > ### Comment · Reviewer_D8FE · 2024-11-27
> > **Response to Authors**
> >
> > Thanks for the detailed response, which helps a lot to better understand this work. I don't have additional questions regarding the work.
> >
> > Below are just some discussions that won't affect my evaluation, and feel free to response with your ideas.
> >
> > > The “zero-shot” claim
> >
> > Thank you for the clarification. I think including this explanation in the revised paper would be helpful. Many terminologies related to LLMs, such as "zero-shot" and "agents," are overused. One of my concerns is whether a "few-shot prompt" can truly be considered "zero-shot" learning, especially given the connection between in-context learning (ICL) and gradient descent in prior works [1,2]. While I understand that this is not an issue specific to your work, I would advocate for careful use of the term "zero-shot."
> >
> > [1]: What learning algorithm is in-context learning? investigations with linear models.
> >
> > [2]: Why can GPT learn in-context? language models secretly perform gradient descent as meta-optimizers
> >
> > > However, given the Markovian property of the studied systems, the full trajectory is not needed in theory, as the following holds:
> > >
> > > $P(s_t|a_{t-1},s_{t-1},...,s_1,a_0,s_0) = P(s_t|a_{t-1},s_{t-1})$
> >
> > Under the assumption of Markovian property, does your proposed ICL-based method achieve the similar results for both $P(s_t|a_{t-1},s_{t-1},...,s_1,a_0,s_0)$ and  $P(s_t|a_{t-1},s_{t-1})$ ?  If not, could this be a double standard between the ICL-based method and MLP approaches? I believe the cold-start problem could pose a potential issue for this work.

---

> > > ### Author Response · Authors · 2024-12-02
> > > **Answer to Reviewer D8FE**
> > >
> > > We are grateful to Reviewer D8FE for their engagement and valuable feedback, which has significantly contributed to improving our work.
> > >
> > > >The “zero-shot” claim
> > >
> > > Regarding the “zero-shot” claim, we fully acknowledge the reviewer's point about the careful use of such terms in machine learning research. In response to the reviewer's recommendations, we have added a paragraph at the end of Section 3.2 to explicitly clarify our stance on the use of the term "zero-shot." We hope that this modification addresses any potential misunderstandings, as suggested by the reviewer.
> > >
> > > >I believe the cold-start problem could pose a potential issue for this work.
> > >
> > > We appreciate the reviewer's insightful question regarding the cold-start problem and potential methodological inconsistencies in the comparison with the MLP baseline.
> > >
> > > Our ICL-based method leverages the attention mechanism's ability to dynamically generate context-dependent layer weights at inference time. While the MLP does not see the samples at inference time, it still saw the same data at training time and used them to update its parameters via gradient descent. From this perspective, we believe the comparison is fair, as both models use the same data (transitions within a given trajectory) to update their weights.
> > >
> > > This is also analogous to how Gaussian Processes (GPs) work with a fixed kernel, where the model's predictive power arises from conditioning on historical data and updating the predictive (posterior) distribution through Bayes' theorem.
> > >
> > > We agree with the reviewer that the two approaches—classical gradient-descent-based learning and in-context learning—are fundamentally different. Designing evaluation methodologies that are fair from all perspectives is a non-trivial task. We believe we have made our best effort to ensure a fair comparison between our method and the baselines, and we remain open to any suggestions from the reviewer to improve our evaluation protocol.

---

### Official Review · Reviewer_1htQ · 2024-11-04

**Soundness:** 3
**Presentation:** 2
**Contribution:** 3
**Rating:** 8
**Confidence:** 3

**Summary:**

The paper explores the utilization of LLMs in the realm of RL, specifically targeting continuous state spaces that have been understudied in the context of LLM integration. It introduces a novel approach termed Disentangled In-Context Learning (DICL), designed to leverage pre-trained LLMs for predicting the dynamics of continuous Markov decision processes. The paper is supported by theoretical analysis and extensive experiments, demonstrating the potential of LLMs in model-based policy evaluation and data-augmented off-policy RL, and shows that LLMs can produce well-calibrated uncertainty estimates.

**Strengths:**

1. The paper presents a method to integrate state dimension interdependence and action information into in-context trajectories within RL environments, enhancing the applicability of LLMs in continuous state spaces.
2. It provides a theoretical analysis of the policy evaluation algorithm resulting from multi-branch rollouts with LLM-based dynamics models, leading to a novel return bound that enhances understanding in this area.
3. The paper offers empirical evidence supporting the benefits of LLM modeling in two RL applications: policy evaluation and data-augmented offline RL, showcasing the practicality of the proposed methods.
4. It demonstrates that LLMs can act as reliable uncertainty estimators, a desirable trait for MBRL algorithms.

**Weaknesses:**

1. The paper does not extensively discuss how the proposed method generalizes across different environments or tasks, that is, more discussion about the application of this method is needed.
2. While DICL simplifies certain aspects of RL, the integration of actions and the handling of multivariate data present ongoing challenges. More discussion about the introduced aspects of the DICL is needed.
3. The experiments are somewhat simplistic, and it would be worthwhile to conduct more in-depth analyses, such as discussing when they become ineffective and more results will be beneficial.
4. The writing style of the paper is a bit convoluted, making it less fluent to read.

**Questions:**

Why does Principal Component Analysis (PCA) decouple features, isn’t it primarily for feature selection? If it is for feature selection, are there any special constraints needed for article-related scenarios?

---

> ### Author Response · Authors · 2024-11-19
> **Answer to Reviewer 1htQ (Part 1/2)**
>
> We thank the reviewer 1htQ for their positive and very encouraging feedback. We are grateful that the reviewer acknolwedged the novelty, theoretical foundations and the benefits of our work to RL.
>
> We address the reviewer's concerns point by point below.
>
> > 1. The paper does not extensively discuss how the proposed method generalizes across different environments or tasks, that is, more discussion about the application of this method is needed.
>
> We acknowledge the Reviewer's concern regarding the diversity of the considered environments in our approach. As stated in the contribution section of the paper, our primary focus was to provide a proof-of-concept of our method in addressing proprioceptive control environments. This has been achieved with the DICL-SAC algorithm on Pendulum, HalfCheetah and Hopper and with policy evaluation on HalfCheetah and Hopper.
>
> To further address Reviewer 1htQ's concern, we include in the revised paper the Table 1 (p.6) that presents an ablation study using different LLMs within DICL, averaging the results over 7 tasks from the D4RL Benchmark: HalfCheetah (random, medium, expert), Hopper (medium, expert), and Walker2d (medium, expert). The details of these experiments are provided in Appendix H.
>
> >2. While DICL simplifies certain aspects of RL, the integration of actions and the handling of multivariate data present ongoing challenges. More discussion about the introduced aspects of the DICL is needed.
>
> The primary goal of DICL is to address the multivariate nature of states and actions in MBRL, as highlighted by Reviewer 1htQ. To enhance the clarity of our method in the revised submission, we have included a figure that illustrates the complete DICL framework (Figure 1). This figure details the process from raw states and actions to final predictions, including the intermediate steps of PCA and in-context learning through the LLM. Additionally, we have provided a concrete prompt example in the background section of the paper (p.3). Lastly, we have introduced a "Discussion" section to explore some future directions that emanate from our method.
>
> >3. The experiments are somewhat simplistic, and it would be worthwhile to conduct more in-depth analyses, such as discussing when they become ineffective and more results will be beneficial.
>
> As shown in the DICL-SAC experiments section, the algorithm is sensitive to the proportion of LLM data used to augment SAC's replay buffer. This is also backed by Theorem 4.2 where the theoretical bound gets tighter when $p$ is small. Therefore, the $\alpha$ hyperparamter of the DICL-SAC algorithm has to be carefully tuned to benefit SAC from the LLM-based data augmentation.
>
> In the revised paper, we also added a *Discussion* section (p.10) to discuss the computational cost of using LLMs, a current limitation of the approach.
>
> >4. The writing style of the paper is a bit convoluted, making it less fluent to read.
>
> We thank the reviewer for their feedback that enabled us to improve the writing and clarity of our work.  We refer the reviewer to the general comment for a summary of the changes made. Besides these changes, we have also modified the writing style and sentence composition at various places in the paper.
>
> These revisions aim to address the reviewer's concerns and provide a clearer and more coherent presentation of our work.

---

> ### Author Response · Authors · 2024-11-19
> **Answer to Reviewer 1htQ (Part 2/2)**
>
> >5. Why does Principal Component Analysis (PCA) decouple features, isn’t it primarily for feature selection? If it is for feature selection, are there any special constraints needed for article-related scenarios?
>
> Although PCA might be primarily known for feature selection and dimension reduction, the main interest of using it in DICL is that it leads to linearly uncorrelated components. Indeed, as DICL applies ICL independently on each feature dimension we need a transformation that decouples the feature dimensions.
> PCA does not necessarily lead to independent components, only linearly uncorrelated features, but appears to be good enough in our experiments. More specifically, PCA captures the linear dependencies between the state (and action) features. This means that if a feature (one of the state or action dimensions) vector can be expressed as a linear combination of the others: $\exists i, f_i \in \text{span}(\{f_j\}_{j \neq i})$, applying the rotation matrix obtained by the PCA projects the data in a space of linearly uncorrelated features: $\forall i,j, i \neq j: \mathbb{E}[(Rf)_i(Rf)_j] = 0$, where $R \in \mathbb{R}^{d \times d}$ is the rotation matrix obtained by PCA and $f \in \mathbb{R}^d$ is the feature vector. Moreover, the features are ordered by their variance: $\forall i < j: \mathbb{E}[(Rf)_i^2] \geq \mathbb{E}[(Rf)_j^2]$. As illustrated in the covariance matrix in Figure 2, many state and action dimensions exhibit strong linear correlations.
>
> Furthermore, as PCA can be used for feature selection, this is an additional advantage of our method. As outlined in Section 3.3, we leverage the dimensionality reduction facilitated by PCA and retain a number of components equivalent to half the number of the original features (which correspond to >85% of explained variance). This idea helps in reducing the overall computational time of the DICL method, as demonstrated in Figure 3.c.
>
> We thank again the reviewer for their valuable feedback. We hope that all the reviewer's concerns and questions have been addressed. We will be happy to answer any additional questions.

---

> ### Author Response · Authors · 2024-11-25
> **Follow-up**
>
> We once again thank reviewer 1htQ for their insightful feedback, which has helped to enhance the quality of the paper. We hope our reply has sufficiently addressed their comments. As the author-reviewer discussion period is ending, we would be glad to read their reply and address any additional questions.

---

> ### Comment · Reviewer_1htQ · 2024-11-26
>
> Thanks for the elaboration of the authors. I think the replies address most of my concerns. Since my initial rating has reflected the core value of this work, I will keep my rating unchanged.

---

> > ### Author Response · Authors · 2024-12-02
> > **Answer to Reviewer 1htQ**
> >
> > We sincerely thank the reviewer for their thorough evaluation and constructive feedback.

---

### Official Review · Reviewer_Um7G · 2024-11-04

**Soundness:** 3
**Presentation:** 2
**Contribution:** 3
**Rating:** 5
**Confidence:** 3

**Summary:**

This paper proposes a new approach Disentangled In-Context Learning (DICL) to generalize LLM-based in-context learning to the domain of continuous-state-space reinforcement learning. This paper then analyzed the theoretical properties of this new DICL approach, and reports empirical experiments results from policy evaluation and data-augmented off-policy RL to the advantages in sample efficiency and performance of DICL in comparison to baseline methods.

**Strengths:**

This paper has good originality in that it innovatively proposes a novel DICL method to generalize ICL to continuous-state-space RL. This paper also includes solid mathematical derivations and proofs and detailed experiment results to support the claims in the paper.

**Weaknesses:**

There is a lot of room of improvement for the clarity, writing and presentation of this paper. Multiple places in the paper are not very clearly explained and the general theme of the paper is a little bit hard to follow in its writing. For example, throughout the whole paper, there is no explicit explanation or demonstrations on how exactly the LLM prompts for DICL are constructed. One or more concrete prompt examples would be very helpful in the paper to help readers understand the core technical details of the proposed DICL method.

This paper has very good potential, but its current form could benefit a lot from a systematic revision that improves its clarity and presentation.

**Questions:**

1. Typo - on Line 63, should it be ‘deferred’ instead of ‘differed’?
2. Typo - on Line 322, should it be ‘The goal is to improve …’?
3. In the DICL-SAC algorithm, what would be the optimal value for \alpha? Are there any intuitions for choosing the optimal \alpha value?

---

> ### Author Response · Authors · 2024-11-19
> **Answer to Reviewer Um7G**
>
> We thank the reviewer Um7G for their detailed and encouraging feedback. We appreciate that the reviewer found our work original and innovative with solid theoretical justifications and detailed experiments supporting our claims.
>
> We address the reviewer's concerns point by point below.
>
> > 1. There is significant room for improvement in the clarity, writing, and presentation of this paper. Several sections are not clearly explained, making the overall theme difficult to follow. For instance, the paper lacks explicit explanations or demonstrations of how the LLM prompts for DICL are constructed. Including one or more concrete prompt examples would greatly aid readers in understanding the core technical details of the proposed DICL method.
>
> To enhance clarity and understanding, we have made the following revisions in the revised version of the paper (as mentioned in the general comment):
>
> - **Figure 1**: A figure explaining the full DICL framework, accompanied by a detailed description in the introduction section.
> - **Prompt Example**: Additional details and a concrete prompt example have been included in the ICL paragraph of the Background section (Section 2).
> - **Ablation Study on the LLM**: In Section 3.4, we have added a table presenting an ablation study on the choice of Llama model. The table includes metrics (MSE and KS) averaged over seven different tasks from the D4RL benchmark.
> - **LLM Used in Experiments**: For each experiment (figure), we have specified the LLM used in the experiment within the figure caption.
> - **Discussion**: A new discussion section has been added to explore additional aspects and future directions related to our work.
>
> Regarding the specific example of LLM prompt within the DICL framework, the pseudo-code outlined in Algorithm 1 (p. 3) describes the prompt construction procedure. This algorithm demonstrates how a numerical-valued univariate time series is tokenized (through rescaling and encoding with precision $k$) and fed to the LLM: "$s_1^1 s_1^2 \dots s_1^k, s_2^1 s_2^2 \dots s_2^k,\dots$" where $s_t^i$ represents the $i$-th digit in the encoding of the time series value at timestep $t$. In DICL, the time series fed to the LLM in this manner are projections of the original multivariate series in the PCA components space. To clarify this in the paper, as suggested by Reviewer Um7G, we have included a detailed illustrative figure of the DICL process (Figure 1), and a more detailed explanation under the in-context learning paragraph of the background section (l133-151).
>
> While we currently don't use textual information as part of the prompt, we thank the reviewer for this suggestion! We agree that such information, when carefully incorporated into the context, can further enhance the results. This is the case in time series forecasting where textual prompts enhance ICL's performance [1]. We believe that our work lays out a foundation for future works to further extend it with such techniques.
>
> > 2. Typo - on Line 63, should it be ‘deferred’ instead of ‘differed’?
> > 3. Typo - on Line 322, should it be ‘The goal is to improve …’?
>
> We thank Reviewer Um7G for identifying these typos, which have been corrected in the revised submission.
>
> > 4. In the DICL-SAC algorithm, what would be the optimal value for $\alpha$? Are there any intuitions for choosing the optimal $\alpha$ value?
>
> The $\alpha$ parameter in the DICL-SAC algorithm represents the proportion of LLM-generated transitions used to train SAC at each time step. Controlling $\alpha$ is directly linked to controlling the $p$ parameter in Theorem 4.2. When more LLM-generated transitions are sampled (higher $\alpha$), a given state in an already visited trajectory has a higher probability of belonging to an LLM branch (higher $p$) since we uniformly sample the in-context trajectories from the replay buffer. Increasing $\alpha$ can enhance the sample efficiency of the underlying RL algorithm by providing more data. However, a larger $\alpha$ also results in a looser bound in Theorem 4.2 which makes it more challenging to achieve policy improvement on the actual environment. Thus, a careful hyperparameter search for small values of $\alpha$ is recommended for an optimal trade-off in the DICL-SAC algorithm. Our experimental results confirm this, with $\alpha=5$% performing best.
>
> We hope that all the reviewer's concerns and questions have been addressed and we remain open to future discussions. Given our explanations and revisions to the paper, we would be grateful if the reviewer could reconsider the evaluation of our work accordingly.
>
> ---
> References:
> - [1] Jin, Ming, et al. "Time-llm: Time series forecasting by reprogramming large language models." arXiv preprint arXiv:2310.01728 (2023).

---

> ### Author Response · Authors · 2024-11-25
> **Follow-up**
>
> We once again thank reviewer Um7G for their insightful feedback, which has helped to enhance the quality of the paper. We hope our reply has sufficiently addressed their comments. As the author-reviewer discussion period is ending, we would be glad to read their reply and address any additional questions.

---

### Author Response · Authors · 2024-11-19
**General Comment**

We thank all the reviewers for thoroughly and carefully reading our paper. We are happy to hear that they found our approach **original and innovative** (Reviewers Um7G, 1htQ, D8FE, fXTe) with **solid theoretical foundation and analysis** (Reviewers 1htQ, D8FE, fXTe) and **extensive empirical results** showcasing our method's **benefits in Reinforcement Learning** (Reviewers Um7G, 1htQ, fXTe).

Besides responding to each reviewer individually, we upload a revised submission of our paper to tackle their concerns. In summary, the revised paper incorporates the following updates, all of which are highlighted in blue for the reviewers' ease of reference.

**Updates on the PDF**:
- **Figure 1**: A figure explaining the full DICL framework, accompanied by a detailed description in the introduction section.
- **Prompt Example**: Additional details and a concrete prompt example have been included in the ICL paragraph of the Background section (Section 2).
- **Ablation Study on the LLM**: In Section 3.4, we have added a table presenting an ablation study on the choice of the Llama model. The table includes metrics (MSE and KS) averaged over seven different tasks from the D4RL benchmark.
- **LLM Used in Experiments**: For each experiment, we have specified the LLM used in the experiment within the figure caption.
- **Discussion**: A new discussion section has been added to explore additional aspects and future directions related to our work.
- **Zero-shot clarification**: Following Reviewer D8FE's recommendation, we have added a paragraph at the end of Section 3.2 to explain our use of the term 'zero-shot' and contextualize it within the relevant literature.

These revisions, along with an improved writing style, have been made to address the valuable feedback provided by the reviewers and to enhance the clarity, writing, and depth of the paper.

We believe that the paper strongly benefited from the reviews, and we respectfully invite the reviewers to reconsider their evaluation score if they think we addressed their concerns.

---

### Meta-Review · Area_Chair_oaD4 · 2024-12-19

**Metareview:**

This work explores using pre-trained LLMs to predict the dynamics of continuous MDPs in-context, addressing challenges related to multivariate data and control signal integration. The reviewers appreciate the originality of the approach, the theoretical analysis, and the thorough empirical evaluation. They also raise concerns, however, primarily on the clarity of presentation, on a few issues with evaluation, and limited discussion on generalizability. The authors respond to all concerns and questions in detail but not all reviewers engage in discussion.

**Additional Comments On Reviewer Discussion:**

The authors respond to the reviewers' concerns and questions, conduct a new ablation, and revise the paper for clarity and to add a discussion section. Overall I feel that the author responses adequately address the majority of the concerns.

---

### Decision · Program_Chairs · 2025-01-22

Accept (Poster)